# A mitotic kinase scaffold depleted in testicular seminomas impacts spindle orientation in germ line stem cells

Heidi Hehnly[1,2†], David Canton[1†], Paula Bucko[1], Lorene K Langeberg[1], Leah Ogier[1], Irwin Gelman[3], L Fernando Santana[4], Linda Wordeman[4], John D Scott[1]*

[1]Department of Pharmacology, Howard Hughes Medical Institute, University of Washington, Seattle, United States; [2]Department of Cell and Developmental Biology, State University of New York Upstate Medical University, Syracuse, United States; [3]Department of Cancer Genetics, Roswell Park Cancer Institute, Buffalo, United States; [4]Department of Physiology and Biophysics, University of Washington, Seattle, United States

**Abstract** Correct orientation of the mitotic spindle in stem cells underlies organogenesis. Spindle abnormalities correlate with cancer progression in germ line-derived tumors. We discover a macromolecular complex between the scaffolding protein Gravin/AKAP12 and the mitotic kinases, Aurora A and Plk1, that is down regulated in human seminoma. Depletion of Gravin correlates with an increased mitotic index and disorganization of seminiferous tubules. Biochemical, super-resolution imaging, and enzymology approaches establish that this Gravin scaffold accumulates at the mother spindle pole during metaphase. Manipulating elements of the Gravin-Aurora A-Plk1 axis prompts mitotic delay and prevents appropriate assembly of astral microtubules to promote spindle misorientation. These pathological responses are conserved in seminiferous tubules from *Gravin*[−/−] mice where an overabundance of Oct3/4 positive germ line stem cells displays randomized orientation of mitotic spindles. Thus, we propose that Gravin-mediated recruitment of Aurora A and Plk1 to the mother (oldest) spindle pole contributes to the fidelity of symmetric cell division.

*For correspondence: scottjdw@u.washington.edu

†These authors contributed equally to this work

Competing interests: The authors declare that no competing interests exist.

## Introduction

Mitotic cell division is a process whereby genetic material is duplicated, separated, and packaged to yield two daughter cells (*Nigg and Raff, 2009*). This process relies heavily on the spatial and temporal synchronization of protein kinase activity at the mitotic spindle, a macromolecular machine that segregates the chromosomes and guides them towards the daughter cells (*Lowery et al., 2004*; *Nigg and Stearns, 2011*; *Langeberg and Scott, 2015*). Correct orientation of the mitotic spindle during cell division combined with local kinase signaling is crucial for cell fate determination, tissue organization, and development (*Yamashita et al., 2007*; *Lesage et al., 2010*; *Gillies and Cabernard, 2011*; *Kiyomitsu and Cheeseman, 2012*; *Pelletier and Yamashita, 2012*; *Joukov et al., 2014*).

The mitotic spindle is constrained by two spindle poles that nucleate microtubules. The mother spindle pole contains the oldest centriole and remains anchored near the stem-cell niche, while the daughter spindle pole migrates to the opposite side of the cell to complete spindle formation (*Yamashita et al., 2007*; *Izumi and Kaneko, 2012*). Recently, a spindle orientation complex has been identified at the mother spindle pole containing protein components that promote maturation (*Yamashita et al., 2007*; *Izumi and Kaneko, 2012*; *Chen et al., 2014*). Several disease-linked genes encode these proteins and their loss causes mitotic delays and spindle misorientation phenotypes (*Buchman et al., 2010*; *Gruber et al., 2011*; *Tan et al., 2014*; *Chen et al., 2014*; *Kim and Rhee, 2014*).

**eLife digest** The genetic material inside our cells is contained within structures called chromosomes. When a cell divides, these chromosomes are copied and then must be correctly divided between the two daughter cells so that each cell has a complete set of genetic material. The correct separation of the chromosomes depends on a structure called the mitotic spindle whose location in the cell also determines where the point of division will be.

Two structures called centrioles are associated with the mitotic spindle and help to organize and direct cell division. The cell carefully controls how these structures are inherited by the daughter cells. For example, when a stem cell divides to produce one stem cell and one cell of a different type, the older centriole can be inherited by the new stem cell. Incorrect placement of the spindle can disrupt this process and is linked to the progression of cancers that affect reproductive organs, such as testicular seminomas.

Here, Hehnly, Canton et al. used biochemical and microscopy techniques to study how the spindle is positioned in cells from patients with testicular seminoma tumors. The experiments reveal that a protein called Gravin is found in lower amounts in the tumor cells than in normal cells. In mice that lack Gravin, the cells in a region of the testicle called the seminiferous tubule divide more rapidly, which is a hallmark of cancer. Gravin accumulates at the end of the spindle where the older centriole is present. This protein acts as a scaffold that holds two enzymes called kinases that regulate cell division in place at the end of the spindle. In the stem cells of the testicle, these kinases also appear to help to correctly position the spindle by organizing the proteins that anchor this end of the spindle to the membrane.

Hehnly, Canton et al.'s findings suggest that Gravin helps to guard against errors occurring during cell division by recruiting two particular kinase enzymes to the mitotic spindle. A future challenge will be to identify the proteins that these kinases affect while anchored to the spindle.

Spindle orientation defects that promote an imbalance between symmetric and asymmetric cell divisions have been implicated in the progression of germ line-derived cancers such as teratomas, seminomas, and ovarian carcinomas (*Neumüller and Knoblich, 2009*). These cancers can be exacerbated by mislocalization or misregulation of mitogenic and mitotic protein kinase cascades (*Carnegie et al., 2009*; *Scott and Pawson, 2009*). The A-kinase anchoring protein Gravin/AKAP12/ SSeCKS has been implicated in the control of mitotic progression (*Xia et al., 2001*; *Gelman, 2010*; *Canton et al., 2012*; *Canton and Scott, 2013*). We now report that Gravin is depleted in proliferating germ line-derived tumors from several patients diagnosed with testicular seminoma. Mechanistic studies show that Gravin is required to spatially coordinate the activities of Aurora A and polo-like kinase 1 (Plk1), two kinases that act in concert to promote spindle orientation.

## Results

### Depletion of Gravin is linked to proliferation in germ line-derived tumor samples (seminomas)

Mutation or amplification of Gravin has been linked to melanoma, prostate, and ovarian cancers, yet nothing is known about the role of this kinase-anchoring protein in solid tumors (*Xia et al., 2001*; *Bateman et al., 2015*; *Finger et al., 2015*). Testicular germ line tumors are the most frequently diagnosed solid cancers in men aged 15–40 years. Currently, 200,000 men develop seminoma annually (*Fung et al., 2007*; *Burum-Auensen et al., 2010*; *Singh et al., 2011*). Although seminoma screening and treatment is well understood, much less is known about the molecular events in germ line stem cells that underlie oncogenesis. Surprisingly, immunoblot analysis of clinical samples from three seminoma patients detected a 9.15-fold reduction in Gravin protein compared to adjacent tissue (*Figure 1A,B*). Interestingly, the loss of Gravin was accompanied by a decrease in two essential cell cycle regulator kinases, Aurora A and Plk1 (*Figure 1A*, mid panels, and *Figure 1B*). Similar trends were observed in four additional clinical samples from seminoma patients (*Figure 1—figure supplement 1A*).

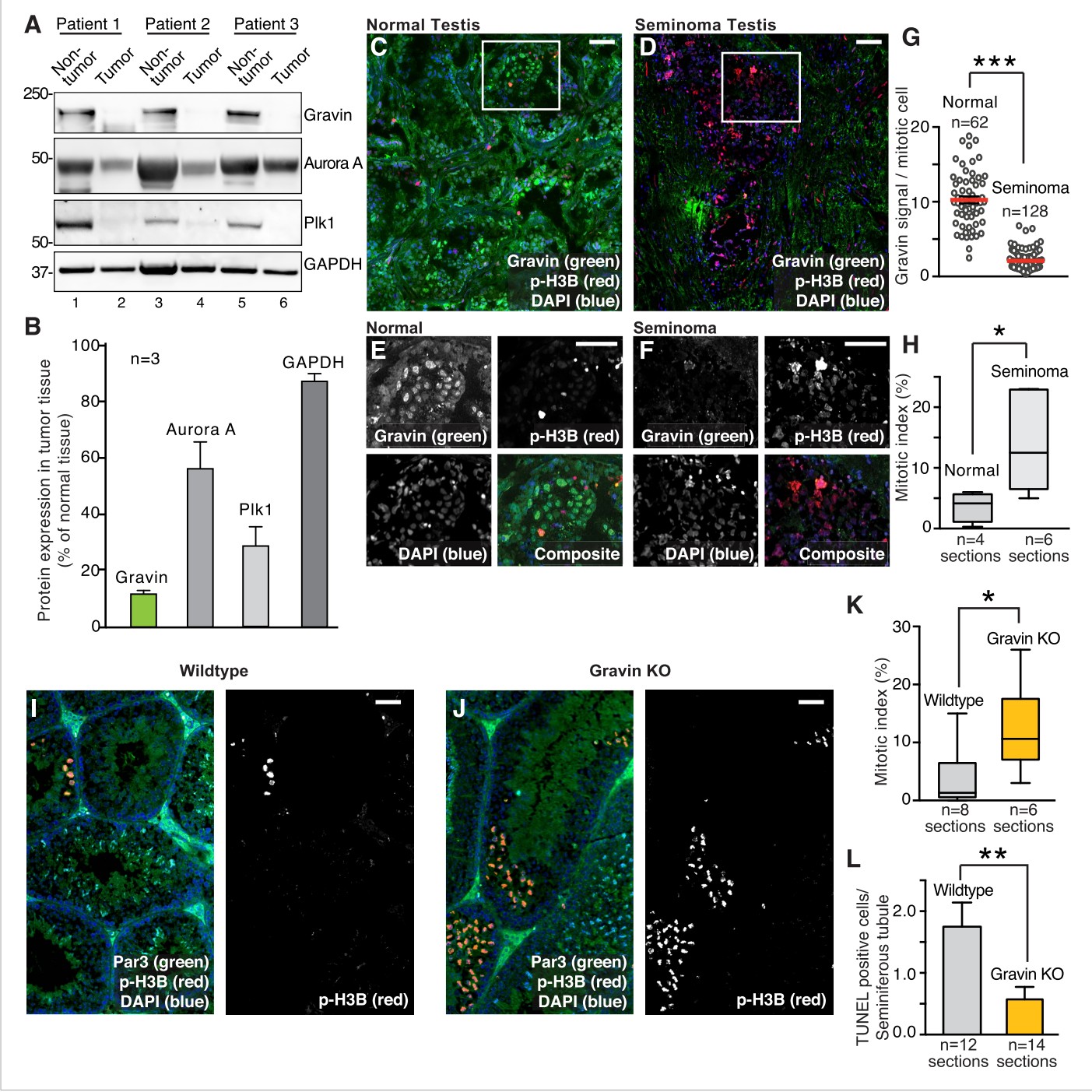

**Figure 1**. Loss of Gravin correlates with perturbed mitosis in human seminomas and mouse seminiferous tubules. (**A**) Immunoblot analysis of tissue lysates from resected seminomas (lanes 2, 4, and 6) and normal adjacent tissue (lanes 1, 3, and 5). Proteins were identified using antibodies against (top) Gravin, (upper-mid) Aurora A, (lower-mid) Plk1, and (bottom) GAPDH loading control. (**B**) Quantification of immunoblot data (**A**) by densitometry (n = 3 ± SEM). (**C**, **D**) Representative testis sections from (**C**) a 30-year-old individual and (**D**) a 26-year-old seminoma patient. Immunofluorescent staining shows Gravin (green), p-H3B (red), and DNA (DAPI, blue). Scale bar, 40 μm. (**E**, **F**) Magnified insets from **C** and **D** are included. Scale bar, 40 μm. (**G**) Gravin signal intensity per mitotic cell was quantified from normal and seminoma sections of testis (p-H3B positive, n-values are indicated, ***p < 0.001). The number of cells used in each analysis is indicated. (**H**) The mitotic index was calculated for (normal; n = 4) and (seminoma; n = 6) tissue sections by determining the percentage of pH3B-positive cells. (*p < 0.05). (**I**, **J**) Related experiments were conducted on testis sections from 7-week-old wild-type (**I**), and Gravin knockout (**J**) mice. Immunostaining with antibodies against Par3 (green), p-H3B (red), and DAPI (blue) is presented. Scale bar, 40 μm. (**K**) Calculation of the mitotic index in testis sections from wild-type (gray) and Gravin knockout (orange) mice. The number of tissue sections measured is indicated below each
*Figure 1. continued on next page*

Figure 1. Continued

column (*p < 0.05). (**L**) TUNEL staining was used to monitor apoptosis in seminiferous tubule sections from wild-type (gray) and Gravin knockout (orange) mice. Data are presented as TUNEL-positive cells per seminiferous tubule. The number of sections is depicted below each column. (**p = 0.01).

The following figure supplement is available for figure 1:

**Figure supplement 1**. Loss of Gravin in human summons and mouse tissues, and correlation with altered mitosis.

Immunofluorescent screening of seminoma tumor sections confirmed a reduction in Gravin (*Figure 1C–G*). In normal testis, Gravin is uniformly distributed throughout all cell types of the seminiferous tubules (*Figure 1C*, green). In contrast, the anchoring protein is regionally distributed in seminoma sections and the fluorescent intensity of the Gravin signal is markedly reduced (*Figure 1D*, green). These differential protein expression patterns are clearly evident in magnified images of the indicated insets (*Figure 1E,F*). Moreover, counterstaining with the mitotic marker phospho-Ser 10-Histone 3B (p-H3B) indicated that a 4.88-fold decrease in Gravin was observed in mitotic cells (*Figure 1E–G*). The mitotic index was elevated in seminoma compared to normal tissue (3.78-fold, *Figure 1H*).

Further evidence that Gravin loss alters mitotic progression was obtained from knockout mice (*Akakura et al., 2008*). Seminiferous tubule sections were stained for a cytoplasmic marker, PAR6 (green), a nuclear marker, DAPI (blue), and the mitotic marker p-H3B (*Figure 1I,J*). Under these conditions, Gravin knockout mice displayed a 3.39-fold increase in the mitotic index compared to wild-type seminiferous tubule sections (*Figure 1I–K*). Gravin knockout mice also exhibited a 3.07-fold reduction in the number of cells undergoing apoptosis (TUNEL-positive cells; *Figure 1L*, *Figure 1—figure supplement 1B,C*). In primary cultures of mouse embryonic fibroblasts (MEFs), Gravin null cells displayed a slower rate of proliferation compared to wild type ([*Akakura et al., 2010*]; *Figure 1—figure supplement 1D*) with a concomitant increase in senescent morphology (*Figure 1—figure supplement 1E*). Collectively, the data in *Figure 1* implicate reduced Gravin expression with changes in cell cycle progression that are observed in germ line-derived solid tumors.

## Phospho-Gravin anchors Aurora A with Plk1 during metaphase

Our initial findings postulate that Gravin loss contributes to the mitotic abnormalities observed in seminoma. One plausible explanation is that loss of Gravin uncouples the location of protein kinases that drive the cell cycle. Phosphorylation of Gravin on Threonine 766 promotes recruitment of Plk1, a kinase that prompts mitotic progression and appropriate spindle formation (*Canton et al., 2012*). A key advance in our studies came with the discovery that Gravin also anchors an upstream mitotic kinase, Aurora A (*Figure 2A*).

To test this concept, four complementary approaches were used. First, immunoblot analysis detected Aurora A and Plk1 in Gravin immune complexes isolated from mitotic cell lysates (*Figure 2A*, lane 6). Control experiments confirmed that the closely related Aurora B kinase does not interact with Gravin (*Figure 2B*, lane 4). Conversely, the RII subunit of protein kinase A (PKA) constitutively interacts with Gravin throughout the cell cycle (*Figure 2A*, lanes 4 and 6). Second, conventional immunofluorescent techniques demonstrate that Aurora A (red), Plk1 (green), and p766-Gravin (blue) are concentrated at mitotic spindle poles (*Figure 2C*). Control experiments confirmed that total Gravin (red) organizes at mitotic spindle poles with a subpopulation dispersed throughout the cell (*Figure 2—figure supplement 1A*), whereas Aurora B localizes to the metaphase plate (*Figure 2—figure supplement 1B*). Third, structured illumination microscopy (SIM, resolution ~100 nm) revealed that p-Gravin, Aurora A, and Plk1 decorated a higher-order lattice-like structure at mitotic spindle poles reminiscent of pericentriolar material (PCM (*Lawo et al., 2012*); *Figure 2D*, *Figure 2—figure supplement 1C*). Lastly, a proximity ligation assay (PLA) was used to pinpoint p766-Gravin association with either kinase during the cell cycle (*Figure 2E–H*). This approach combines antibody recognition with amplification of a DNA hetero-duplex to mark discrete protein–protein interaction pairs that reside within 40–60 nm of each other (*Samelson et al., 2015*). Quantification of PLA puncta indicated that p766-Gravin/Aurora A sub-complexes was enhanced 3.86-fold during metaphase when compared to interphase cells (*Figure 2E,F*). The p766-Gravin-Plk1 sub-complex was

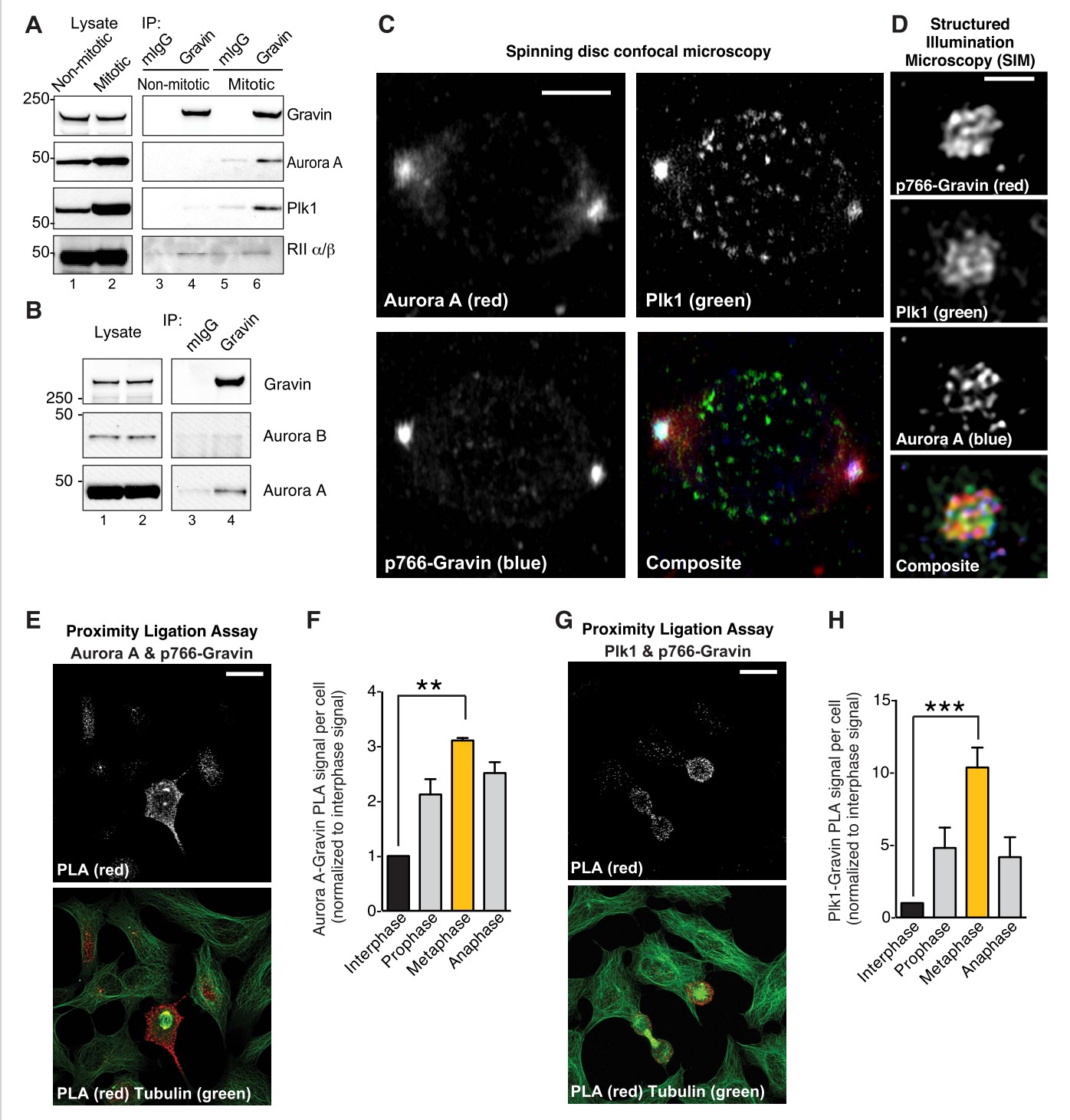

**Figure 2**. Phospho766-Gravin interacts with Plk1 and Aurora A during mitosis. (**A**) Endogenous Gravin complexes were immunoprecipitated from non-mitotic HEK293 cell lysates (lanes 3 and 4) and mitotic cell lysates (lanes 5 and 6). Samples were immunoblotted for (top) Gravin, (upper-mid) Aurora A, (lower-mid) Plk1, and (bottom) RII subunit of protein kinase A (PKA). Control immunoprecipitations (mIgG) are included (lanes 3 and 5). (**B**) Endogenous Gravin complexes were immunoprecipitated from mitotic lysate (lane 4) and immunoblotted for (top) Gravin, (mid) Aurora B, and (bottom) Aurora A. Control immunoprecipitations (mIgG) are included (lane 3). (**C**) Metaphase cells were immunostained for Aurora A (red), Plk1 (green), and p766-Gravin (blue). Confocal micrographs are presented as maximum projections. A composite image is included. Scale bar, 5 µm. (**D**) A structured illumination microscopy (SIM) maximum projection of a single mitotic spindle pole decorated with antibodies to Aurora A (blue), p766-Gravin (red), and Plk1 (green). Scale bar, 1 µm. (**E–H**) A proximity ligation assay (PLA) was used to detect in situ interaction between (**E**, **F**) Aurora A/p766-Gravin and

*Figure 2. continued on next page*

*Figure 2. Continued*

(**G**, **H**) Plk1/p766-Gravin during the cell cycle. (**F**, **H**) The integrated PLA signal intensity per cell was calculated for different stages of the cell cycle. Each value was normalized to the signal obtained in interphase cells (n = 3 experiments ± SEM). Phospho-Gravin interaction with Aurora A (**F**, **\*\*p < 0.001**) and Plk1 (**H**, **\*\*\*p < 0.0005**) was maximal during metaphase as compared to interphase or other phases of the cell cycle.

The following figure supplement is available for figure 2:

**Figure supplement 1**. Subcellular location of Gravin complex components during mitosis.

enriched 10.34-fold (*Figure 2G,H*). Thus, we conclude that p766-Gravin scaffolds Aurora A with Plk1 at a PCM-like structure on mitotic spindle poles during metaphase.

## Clustering Aurora A and Plk1 at spindle poles requires Gravin

Since p766-Gravin, Aurora A, and Plk1 decorate mitotic spindle poles, we reasoned that the anchoring protein may actively constrain both enzymes at this location. To test this hypothesis, the subcellular distribution of both kinases was evaluated in MEFs from wild-type and Gravin knockout mice (*Figure 3A–H*). Metaphase cells were identified by the presence of a bipolar microtubule spindle (*Figure 3A*, top panel, green). Aurora A (red) and Plk1 (blue) were enriched at mitotic spindle poles (*Figure 3A*, top panels); however, both kinases were less evident at mitotic spindle poles of Gravin null MEFs (*Figure 3A*, bottom panels). Signal intensity measurements at mitotic spindle poles provided a quantitative analysis of this phenomenon (*Figure 3B–E*). Gravin null MEFs exhibited a loss of p766-Gravin (*Figure 3B*), a 3.05-fold reduction in Aurora A (*Figure 3C*), and a 1.87-fold reduction in Plk1 (*Figure 3D*) when compared to wild-type MEFs. Signal intensities for the spindle pole marker pericentrin were equivalent in cells from both genotypes (*Figure 3E*).

Aurora A and Plk1 are required for spindle pole maturation (*Joukov et al., 2014*). One aspect of this vital process is the formation of astral microtubules that project out from mitotic spindle poles to the cell cortex to regulate spindle orientation (*Kotak and Gönczy, 2013*). Super-resolution analysis by SIM revealed a near complete loss of astral microtubules protruding from the spindle poles of Gravin null MEFs when compared to wild-type controls (*Figure 3F,G*). This phenomenon is perhaps most clearly demonstrated upon comparison of magnified sections of spindle pole regions from representative cells of both genotypes (*Figure 3F*, right panels).

Mitotic spindle poles are classified as either the mother spindle pole that contains material that formed the original centriole or the more recently assembled daughter spindle pole (*Bornens, 2012*). Upon closer inspection, we noted that p766-Gravin was unequally distributed between the spindle poles of metaphase cells (*Figure 4A*). A technique known as ground state depletion microscopy followed by individual molecule return (GSDIM) was used to rigorously evaluate this phenomenon (*Fölling et al., 2008*). On the basis of signal overlap with the mother centriole marker Cenexin, we were surprised to discover that components of the Gravin kinase scaffold were selectively enriched at this location (*Figure 4B,C*, and *Figure 4—figure supplement 1A–E*). This observation was corroborated by proximity ligation as p766-Gravin/Aurora A PLA puncta were 4.16-fold more prevalent at mother spindle poles than at daughter spindle poles (*Figure 4—figure supplement 1F, G*). Likewise, p766-Gravin/Plk1 PLA pairs were enriched 2.35-fold at the mother spindle pole (*Figure 4—figure supplement 1G*). Thus, Gravin and both kinases are tethered within 20–40 nm of each other and form a locally anchored signaling complex at mother spindle poles.

GSDIM was also employed to test whether the asymmetric distribution of Plk1 to the mother spindle pole required Gravin (*Figure 4D,E*). Surprisingly, the location of Plk1 changed upon shRNA-mediated gene silencing of Gravin (*Figure 4—figure supplement 1H*). Plk1 now predominated at the daughter spindle pole (*Figure 4E,F*). This unexpected finding demonstrates that Gravin governs the preferential recruitment of Aurora A and Plk1 to mother spindle poles, one of their appropriate sites of action during mitosis. Additional control experiments in HEK293 cells revealed that cellular levels of both kinases were unaltered in absence of Gravin (*Figure 4—figure supplement 1H*). This latter observation argues that ablation of Gravin in tissue culture cell lines instigates the displacement of both kinases from the spindle pole but may not affect the cellular levels of each enzyme. However, on

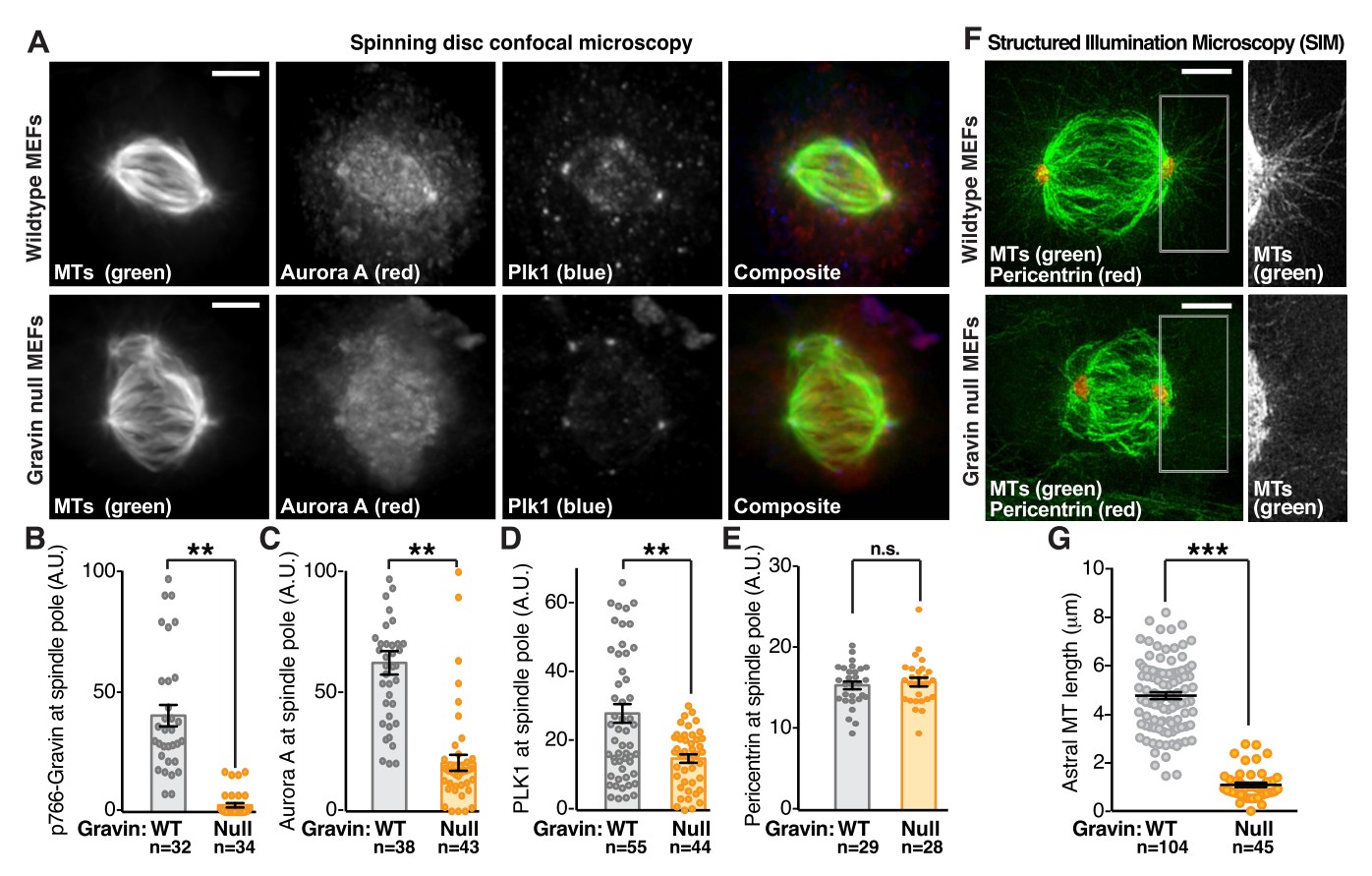

**Figure 3**. Gravin impacts the protrusion of astral microtubules. (**A**) Confocal micrographs of metaphase primary MEFs derived from wild-type (top) and Gravin knockout (bottom) mice are presented as maximum projections. MEFs from each genotype were immunostained for tubulin (MTs, green), Aurora A (red), and Plk1 (blue). Composite images are included. Scale bar, 5 µm. (**B–E**) Quantification of immunofluorescent signal at mitotic spindle poles in wild-type (gray) and Gravin null (orange) MEFs is presented for (**B**) p766-Gravin, (**C**) Aurora A, (**D**) Plk1, and (**E**) a spindle pole marker pericentrin. Total cell numbers used in calculation are indicated below each column. Data are from three independent experiments, **p-values <0.01, ±SEM. (**F**) Astral microtubules are imaged at metaphase using SIM. Maximum projection images of wild-type (top) and Gravin null (bottom) MEFs immunostained with antibodies for tubulin (MTs, green) and pericentrin (red) show microtubules and spindle poles in these cells. Insets depict a magnified view of the astral microtubules protruding from the spindle pole in each genotype. (**G**) Quantitation of astral microtubule (MT) length in wild-type (gray) and Gravin null (orange) metaphase MEFs. Total cell numbers used in calculation are indicated below each column (***p < 0.0001, amalgamated data from three independent experiments).

the basis of our analysis of clinical samples in *Figure 1*, we propose that unidentified mitigating factors contribute to the mislocalization and depletion of both kinases in human tissue.

We reasoned that one consequence of the aforementioned result could be that differential localization of the Gravin scaffold to the mother spindle pole favors more robust assembly of astral microtubules. In order to test this hypothesis, we evaluated astral microtubule abundance, length, and ultrastructure at super-resolution using SIM (*Figure 4F–I*). Pericentrin served as a universal spindle pole marker (*Figure 4F* and *Figure 4—figure supplement 1I*). Centrobin staining was selectively detected daughter spindle poles (*Figure 4F* and *Figure 4—figure supplement 1J*). In wild-type MEFs, we observed protrusion of astral microtubules from both spindle poles (*Figure 4F*). However, upon quantification of three-dimensional (3D) reconstructed SIM images, it was evident that astral microtubules protruding from the mother spindle pool were longer than those emanating from the daughter spindle pole (*Figure 4G*). Next, we investigated this process in Gravin null MEFs to ascertain whether the anchoring protein or its binding partners influence this phenomenon. Notably in Gravin null MEFs, the length of astral microtubules protruding from the mother spindle pool was reduced (*Figure 4H*). Therefore, on the basis of these findings, we can propose that Gravin or elements of its

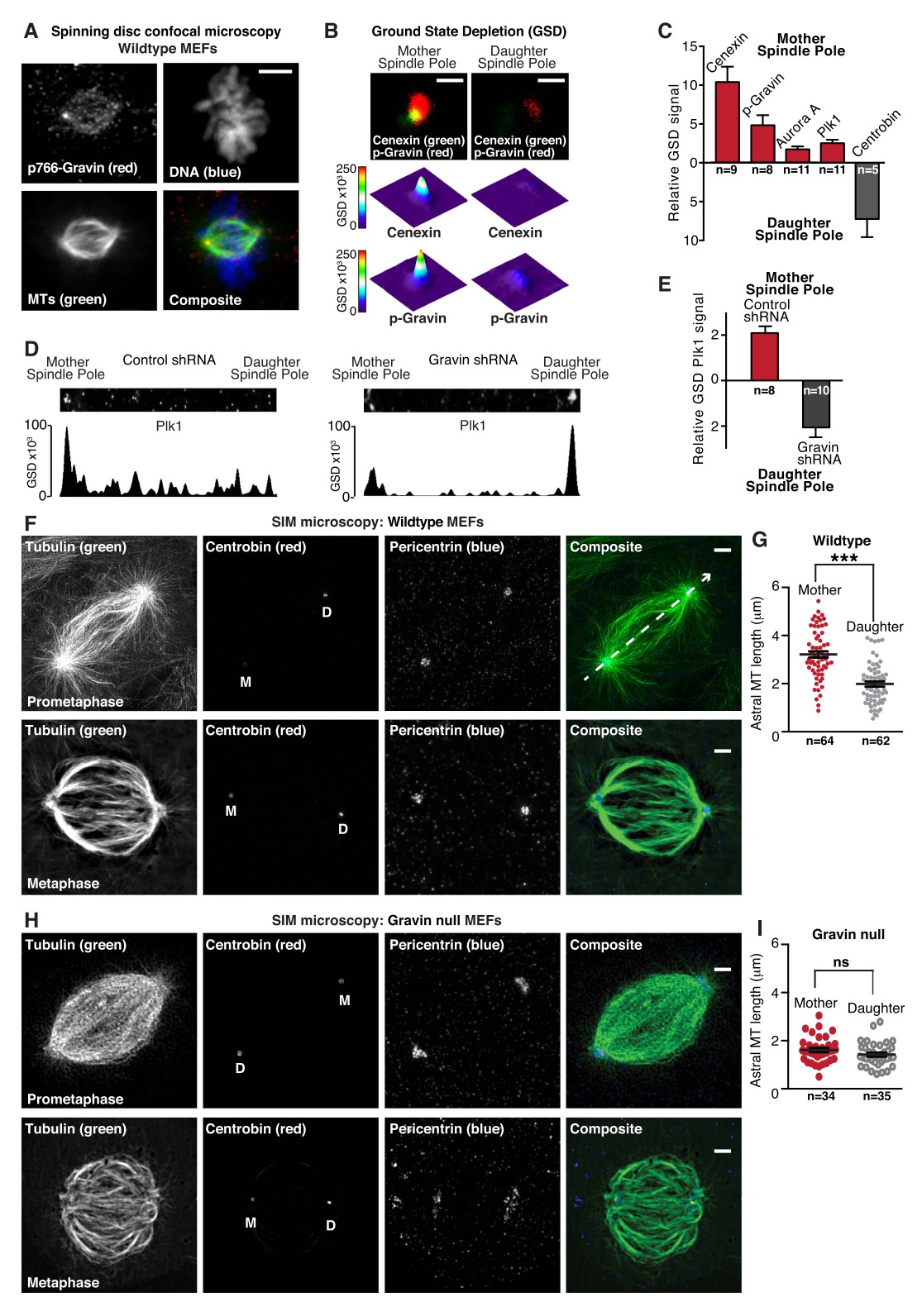

**Figure 4**. Gravin-Aurora A-Plk1 scaffold is preferentially sequestered at mother spindle poles. (**A**) Spinning disc confocal micrograph (maximum projection) of a metaphase wild-type MEF depicts asymmetric enrichment of p766-Gravin (red) at one spindle pole. Counterstaining with tubulin (MTs, green) and DAPI (DNA, blue) are shown. Composite image is shown. Bar, 5 μm. (**B**) Ground state depletion microscopy (GSDIM) was performed on prometaphase HEK293 cells (top). Cells were immunostained for a mother spindle pole marker, Cenexin (green) and p-Gravin (red). Quantification of

*Figure 4. continued on next page*

*Figure 4. Continued*

these signals is shown below micrographs. Integrated intensity profiles for (top) cenexin and (bottom) p-Gravin at the mother spindle pole (left). Intensity profiles for both proteins at the daughter spindle pole are also presented (right). Scale bar, 1 µm. (**C**) Relative GSD signals for p766-Gravin, Aurora A, Plk1, and Cenexin at the mother spindle pole (red). Centrobin (gray) was used as a daughter spindle pole marker. Cell numbers used in each calculation are indicated on graph (n = 3 experiments ± SEM). (**D**) GSDIM micrographs showing the distribution of Plk1 at spindle poles in (top, left) control and (top, right) Gravin-depleted HEK293 cells. Densitometric analyses depict the asymmetric distribution of Plk1 at mother and daughter spindle poles in (bottom, left) control and (bottom, right) Gravin knockdown cells. (**E**) Amalgamated data are shown in graph. Cell numbers used in each calculation are indicated on graph (n = 3 experiments ± SEM). (**F**) SIM maximum projection of (top) wild-type and (bottom) Gravin null MEFs at metaphase. Immunostaining for tubulin (green), centrobin (red), and pericentrin (blue) are presented. The daughter spindle pole was decorated with centrobin (red) and marked on the micrograph with D, whereas the mother spindle pole is denoted with M. Composite images are included. Dashed line (white) depicts path of line-scan used to determine which pole contained the most centrobin (see *Figure 4—figure supplement 4I and J*). Scale bar, 2 µm. (**G**) Comparison of astral microtubule (MT) length (µm) protruding from the mother (red; n = 64) and daughter spindle poles (gray, n = 62) in wildtype MEFs (n = 5 cells, ±SEM, ***p < 0.0001). (**H–I**) Quantitation of astral microtubule (MT) length protruding from the mother (red; n = 34) and daughter spindle poles (gray, n = 35) in Gravin null MEFs (n = 5 cells, ±SEM, ns depicts not significant).

The following figure supplement is available for figure 4:

**Figure supplement 1**. Super-resolution microscopy identifies p-Gravin, Plk1, and Aurora A location and codistribution in mitotic cells.

signaling scaffold contribute to the assembly of astral microtubules. Additional rescue experiments with a Gravin mutant (T766A) that is unable to interact with Plk1 (*Canton et al., 2012*) were achieved at a low frequency but did not restore robust protrusion of astral microtubules (*Figure 4—figure supplement 1K*). Nonetheless, we contend that Gravin-mediated anchoring of Plk1 to the mother spindle pole contributes to the assembly or maintenance of astral microtubules.

## Gravin scaffolds a mitotic kinase cascade

We propose that Aurora A and Plk1 are assembled into a kinase cascade at the mother spindle pole through their simultaneous association with p766-Gravin. Cellular and molecular validation of this model was conducted in three phases. First, Aurora A/Plk1 PLA pairs were localized throughout the cell with a subpopulation accumulating at one mitotic spindle pole (*Figure 5A*). Importantly, a 1.77-fold reduction in the PLA signal was measured upon shRNA depletion of Gravin (*Figure 5B*). This implies that Gravin is necessary to co-localize both kinases.

Second, we tested whether Gravin-associated Aurora A was active. Kinase activity toward a heptapeptide substrate (Kemptide) was measured in Gravin immune complexes isolated from mitotic lysates. Aurora A kinase activity was defined as the number of counts (cpm x $10^3$/IP) blocked by the inhibitor drug VX-680 (*Tyler et al., 2014*). Incubation with VX-680 reduced Kemptide phosphorylation by 33.5% (*Figure 5C*, n = 3). The remaining kinase activity was blocked by the PKA inhibitor peptide PKI and can be attributed to Gravin-associated PKA ([*Lester et al., 1996*]; *Figure 5C*). Third, phospho-peptide antibodies served as an independent index to detect active Aurora A (p-288) and active Plk1 (p-210). Both active kinases were prominent in mitotic lysates from cells treated with control shRNA (*Figure 5D*, lane 3). Gravin-depletion resulted in a twofold reduction in the p288-Aurora A and p210-Plk1 signals as detected by immunoblot (*Figure 5D* lane 4, *Figure 5E*). The active kinases were absent from lysates prepared from interphase cells (*Figure 5D*, lanes 1 and 2). Collectively, these experiments indicate that Gravin constrains active Aurora A and Plk1 to facilitate signal relay from one kinase to the next (*Figure 5F*).

## Gravin anchoring of Aurora A and Plk1 is required for mitotic progression

Mechanistic studies examined whether a Gravin-Aurora A-Plk1 scaffold manages mitotic progression. As a prelude to these studies, it was necessary to generate reagents that displace Aurora A and Plk1 from Gravin. In vitro binding studies using purified GST-Gravin fragments mapped a central region of the anchoring protein (amino acids 451 to 900) that directly interacts with Aurora A (*Figure 6A* top panel, lane 4). More detailed mapping studies defined at least two extended Aurora A-binding sequences within this region, thus eliminating site-directed mutagenesis as the most direct approach to disrupt the Aurora A-Gravin interaction. As an alternate approach, we employed ectopic

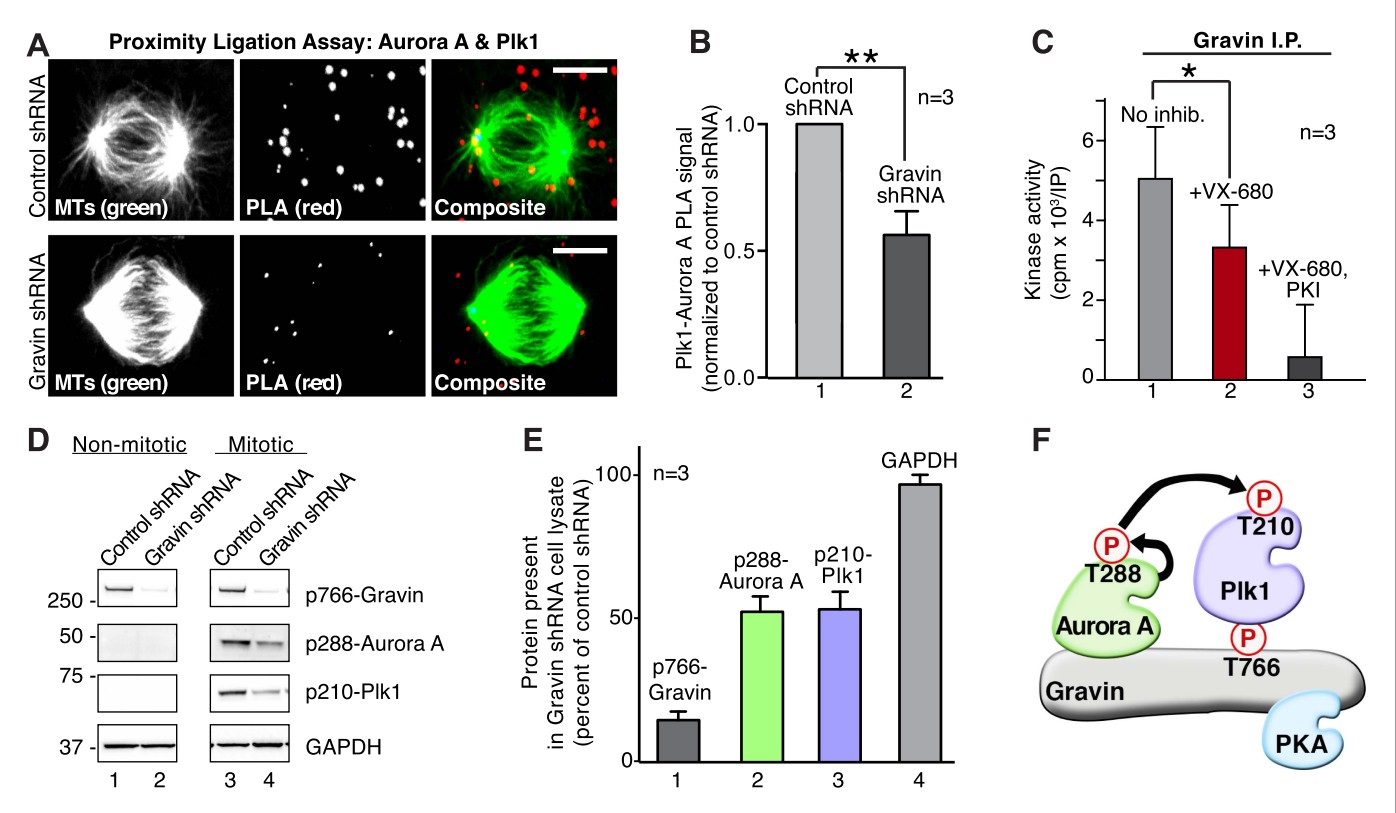

**Figure 5**. Gravin scaffolds an Aurora A and Plk1 kinase-network. (**A**) PLA (red) to identify the frequency and subcellular distribution of the in situ interaction between Aurora A and Plk1 in mitotic HEK293 cells. Staining of microtubules (green) is indicated. (Top) Cells treated with control shRNA, and (bottom) cells treated with Gravin shRNA. Bar, 5μm. (**B**) Quantitation of PLA signal intensity in (light gray) control and (dark gray) Gravin-depleted cells. One hundred fifty cells were analyzed for each condition from three independent experiments (±SEM, **p < 0.01). (**C**) Gravin immune complexes isolated from mitotic lysates were assayed for protein kinase activity using Kemptide (300 μM) as a substrate. Quantitation of $^{32}$P phosphate incorporation was measured by scintillation counting (n = 3 ± SEM, *p < 0.05). Total kinase activity (gray) is compared to enzyme activity in the presence of the Aurora A inhibitor alone (VX-680, red) or VX-680 and a PKA inhibitor (PKI, black). (**D**) Phospho-peptide antibodies were used as an index of Aurora A and Plk1 activity in Gravin immune complexes. Immunoblots show levels of (top) p766-Gravin, (mid) p288-Aurora A, and (lower) p210-Plk1 upon shRNA mediated depletion of Gravin from HEK293 cells. (Bottom) GAPDH loading controls are indicated. (**E**) Densitometric analysis of amalgamated data from three experiments as shown in (**D**) (±SEM). (**F**) A model depicting the proposed flow of phosphorylation signals through a Gravin associated Aurora A and Plk1 cascade.

expression of a Gravin fragment as the primary means of displacing Aurora A from the scaffold. Mutation of Thr766 to Ala in the context of the Gravin 451-900 fragment (*Figure 6—figure supplement 1A*) created a cell-based 'Aurora A disruptor' that antagonizes Aurora A anchoring (*Figure 6B*) without impacting Plk1 (*Canton et al., 2012*). Cells stably expressing histone 2B-Green Fluorescent Protein (GFP) and transfected with the 'Aurora A disruptor' were monitored by live-cell video microscopy (*Figure 6C–F*). A completed cell cycle was defined as the time from chromatin condensation to the initiation of cytokinesis. Expression of the Aurora A disruptor caused a mitotic delay (mean time 85.17 ± 3.77 min, n = 55) when compared to control cells (64.57 ± 4.09 min, n = 95; *Figure 6C,D*). Perhaps, the best depiction of this mitotic defect is upon comparison of time-lapse videos (*Video 1*). Thus, correct targeting of Aurora A facilitates efficient mitosis.

To define the Gravin-Plk1 interface during mitosis, we monitored cells depleted of Gravin and rescued with kinase-binding mutants of the anchoring protein. Metaphase was delayed in Gravin shRNA depleted cells (78.04 ± 3.06 min, n = 29) compared to control (45.86 ± 1.27 min, n = 28, *Figure 6E,F*). This defect was corrected upon rescue with murine Gravin (57.41 ± 3.63 min n = 37; *Figure 6E–G* and *Figure 6—figure supplement 1B*). In contrast, rescue with Gravin T766A, a Plk1-binding defective mutant, was unable to restore normal mitotic progression (90.32 ± 6.33 min, n = 31; *Figure 6E,F,H*). Experiments with a GravinΔPKA mutant that cannot anchor PKA (*Nauert et al., 1997*)

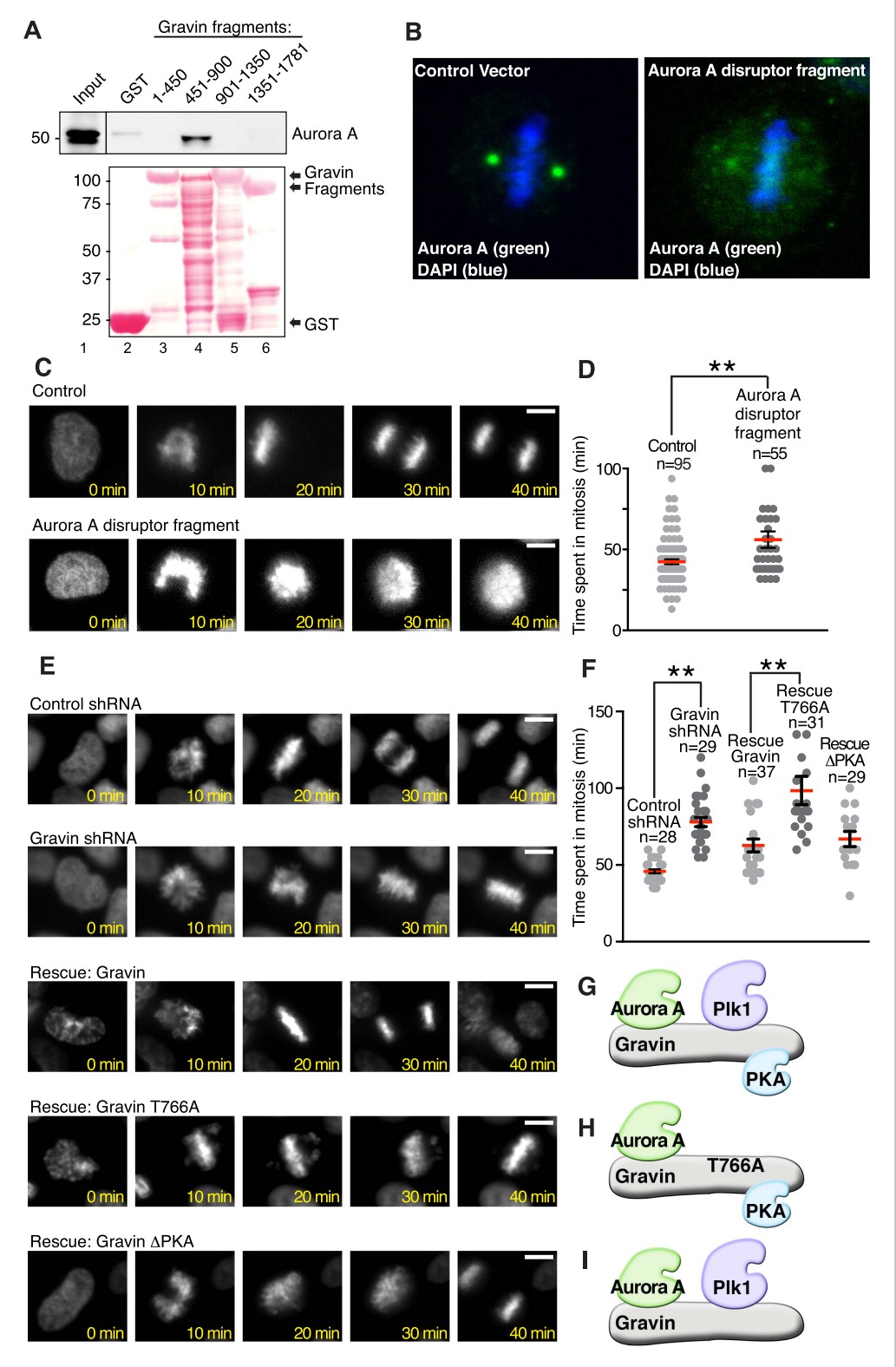

**Figure 6**. Gravin-scaffolding of Aurora A and Plk1 facilitates metaphase progression. (**A**) Direct binding of purified Gravin GST-fusion proteins (first and last amino acid number of each fragment is denoted above each lane) with recombinant V5-tagged Aurora A kinase (generated by in vitro transcription and translation). (Top) Immunoblot detection of Aurora A in complex with GST-Gravin fragments. (Bottom) Ponceau stained blot shows protein expression levels. (**B**) Spinning disc confocal image (maximum projection) of a metaphase cell (control vector, left)
*Figure 6. continued on next page*

*Figure 6. Continued*

stained for (green) Aurora A and (blue) DNA. The subcellular rearrangement of Aurora A following over-expression of the Aurora A disruptor fragment (right). DNA is shown in blue (DAPI). (**C**) Live cell imaging of HeLa cells stably expressing H2B-GFP were monitored through mitosis from the time of DNA condensation until anaphase exit. Shown are (top panels) a representative control cell and (bottom panels) a cell expressing the Aurora A disruptor fragment. Bar, 5 μm. (**D**) Amalgamated data from multiple cells stably expressing H2B-GFP and monitored for time spent in mitosis. Control cells (n = 95 cells) and Aurora A disruptor expressing cells (n = 55 cells) were from three independent experiments (**p < 0.001 ). (**E–I**) Live cell imaging time courses (0–40 min) of cells stably expressing H2B-GFP transfected with (top) control shRNA and (second) Gravin shRNA. Rescue experiments as indicated with (third) murine Gravin; (fourth) murine Gravin T766A; and (fifth) murine GravinΔPKA. Bar, 5 μm. (**F**) Amalgamated data from multiple cells treated with control or Gravin shRNA, and rescued with murine Gravin as shown in **E**. These cells were stably expressing H2B-GFP and monitored for time spent in mitosis. Total cell numbers are indicated on graph (from three independent experiments, **p-values <0.001). (**G–I**) Models depicting the kinase-binding properties of the Gravin mutants used in time course experiments **E** and **F**: rescue with intact Gravin (**G**), Gravin T766A (**H**), and GravinΔPKA (**I**).

The following figure supplement is available for figure 6:

**Figure supplement 1**. Biochemical validation of reagents used in analysis of cell cycle progression.

restored normal cell cycle progression, thereby excluding a role for Gravin-associated-PKA in this process (55.69 ± 3.41 min, n = 29; *Figure 6E,F,I*). Collectively, these studies propose a mechanism where Aurora A proximity to Plk1 facilitates efficient mitotic progression, as both kinases are constrained by Gravin.

## The Gravin macromolecular complex is required for spindle orientation

The orientation of the mitotic spindle during cell division determines whether a polarized epithelium will expand symmetrically (*Gillies and Cabernard, 2011*; *Williams et al., 2014*). Orientation is influenced by the distribution of molecular components at mitotic spindle poles (*Yamashita et al., 2007*; *Lesage et al., 2010*; *Januschke et al., 2011*; *Chen et al., 2014*). Therefore, we reasoned that manipulation of the Gravin scaffold could affect spindle orientation during cell division. To test this hypothesis, we used a spindle tilt assay to measure the angle of each mitotic spindle relative to the substratum ((*Delaval et al., 2011*; *Hehnly and Doxsey, 2014*) *Figure 7A*). Ectopic expression of the Aurora A disruptor perturbed spindle orientation. Mislocalization of Aurora A promoted a 20° misalignment of the mitotic spindle when compared to control cells transfected with empty vector (*Figure 7B,C*). Most control spindles were parallel to the substratum (averaging around 5–10°; *Figure 7D,E*). Yet in Gravin-depleted cells, the perturbation of spindle orientation was 40° (*Figure 7D,E*). Notably, these more severe spindle angle defects were rescued upon expression of the mouse Gravin ortholog (*Figure 7D,E*). In contrast, spindle angle defects were only partially rescued upon expression of the Plk1-binding defective Gravin T766A mutant (angles averaging around 20°; *Figure 7D,E*). The magnitude of this partial rescue is reminiscent of the results obtained upon mislocalization of Aurora A (*Figure 7C*). Further support was provided upon analysis of 3D reconstructed images collected from wild-type and Gravin null MEFs. In wild-type MEFs, the alignment of the mitotic spindle was parallel to the substratum as assessed by the positioning of the spindle poles (*Figure 7F* and *Video 2*, left). In contrast, abnormal orientation and positioning of mitotic spindles were observed in Gravin null MEFs (*Figure 7G* and *Video 2*, right). Thus, we propose that Gravin, through its interaction with Aurora A and Plk1, ensures correct spindle orientation during mitosis.

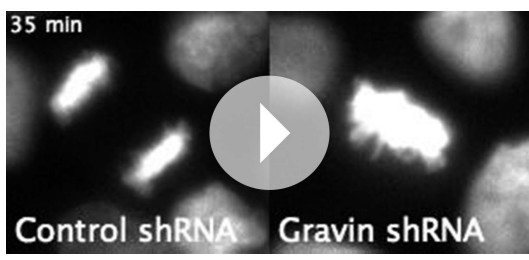

**Video 1.** Comparison of mitotic progression in control and Gravin-depleted cells. Time-lapse video of HeLa cells stably expressing Histone H2B-GFP. Frame by frame comparison of mitotic progression in cells transfected with (left) control shRNA and (right) Gravin shRNA.

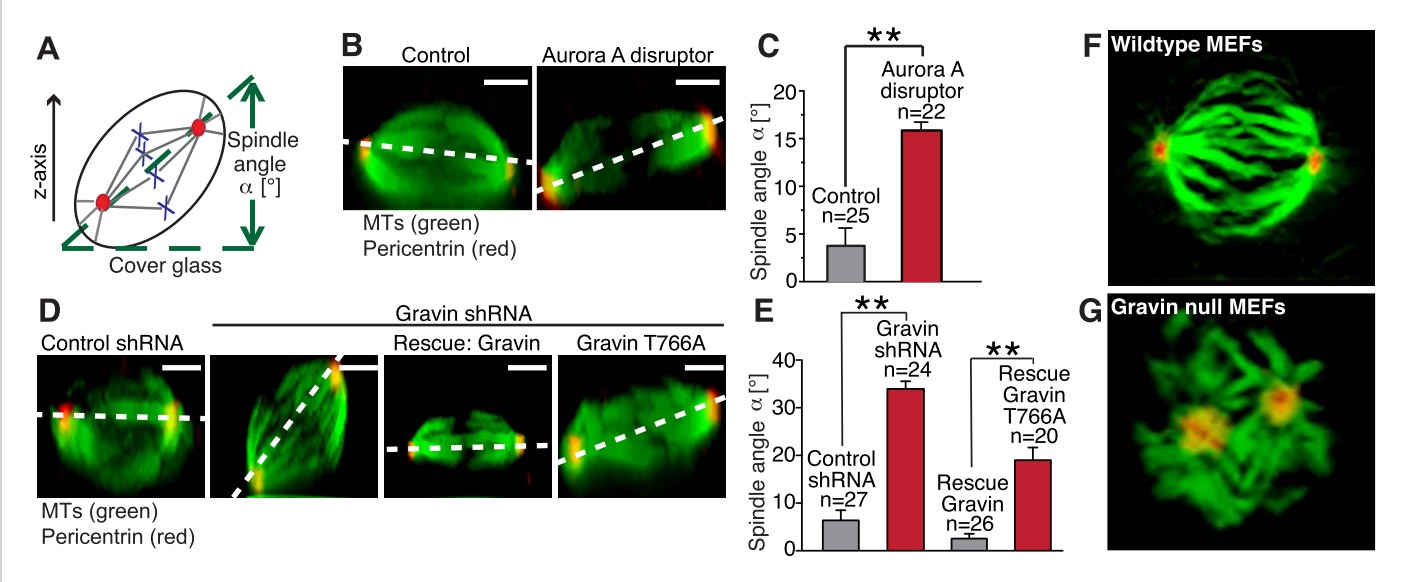

**Figure 7**. The Gravin-Aurora A-Plk1 scaffold regulates appropriate spindle orientation. (**A**) Diagram depicting how spindle angle was calculated for treatments in (**B**–**E**). The z-axis of a metaphase cell with a defined spindle angle α [°] in relation to the cover glass. (**B**) A representative z-axis confocal projection for HeLa cells expressing an empty vector (left) or the Aurora A disruptor fragment. Cells were stained with tubulin (green, MTs) and centrosomes (red, pericentrin). The dashed line connects the two spindle poles and is used to determine the spindle angle relative to the cover glass. Bar denotes 3 μm. (**C**) Spindle angles between cover glass and line bisecting spindle poles in z-axis projections were measured. The amalgamated data from three independent experiments show mean spindle angles ([°]; total cell numbers are denoted above each column, ±SEM, **p-values <0.001). (**D**) A representative z-axis confocal projection for HeLa cells treated with control shRNA and Gravin shRNA and rescue experiments with murine Gravin and the murine Gravin T766A. Cells were stained with tubulin (green, MTs) and centrosomes (red, pericentrin). The dashed line connects the two spindle poles and is used to determine the spindle angle relative to the cover glass. Bar denotes 3 μm. (**E**) Spindle angle quantification for each treatment (cell numbers depicted on graph ± SEM, **p-vales < 0.001). (**F**, **G**) Single frames from z-axis confocal 3-dimensional videos of (**F**) wild-type and (**G**) Gravin null MEFs in metaphase. Staining with tubulin (green) and pericentrin (red). Full videos are presented in *Video 2*.

## Gravin contributes to spindle orientation of germ line stem cells during spermatogenesis

In vivo analyses of seminiferous tubules tested whether the Gravin scaffold determines spindle orientation of germ line stem cells during spermatogenesis. Initially, we examined the morphology of tissue sections from wild-type and Gravin knockout mice. Seminiferous tubules are organized into ascending cellular layers with germ line stem cells (spermatogonia, magenta) residing on the inner face of the basement membrane (*Figure 8A*). Appropriate cellular layering was observed in wild-type sections stained for Gravin (green) and nuclei (white) (*Figure 8B*, left). Parallel sections stained for acetylated tubulin established the correct lumenal organization of elongating spermatids (*Figure 8C*, left). In contrast, the cellular layering of seminiferous tubules and lumenal organization of spermatids are disrupted in Gravin knockout mice (*Figure 8B,C*; right). Of note, the diameter of the lumen is significantly decreased in seminiferous tubules from Gravin knockout mice (*Figure 8D*). In addition, the elongating spermatids are haphazardly dispersed throughout the lumen (*Figure 8C*, right). We postulate that both morphological changes are a consequence of abnormal spindle orientation within dividing germ line stem cells.

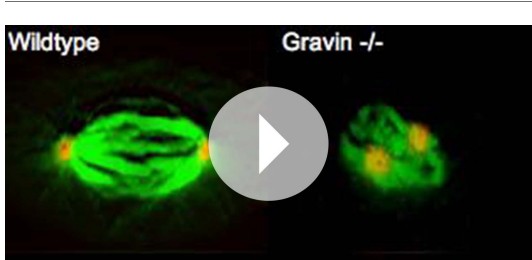

**Video 2.** Comparison of mitotic spindles formed in wild-type and Gravin null MEFs. Reconstructed z-axis confocal 3-dimensional movies mitotic spindles from (left) wild-type and (right) Gravin null MEFs. The mitotic spindles (tubulin, green) and spindle poles (pericentrin, red) are indicated. Related to *Figure 7F,G*.

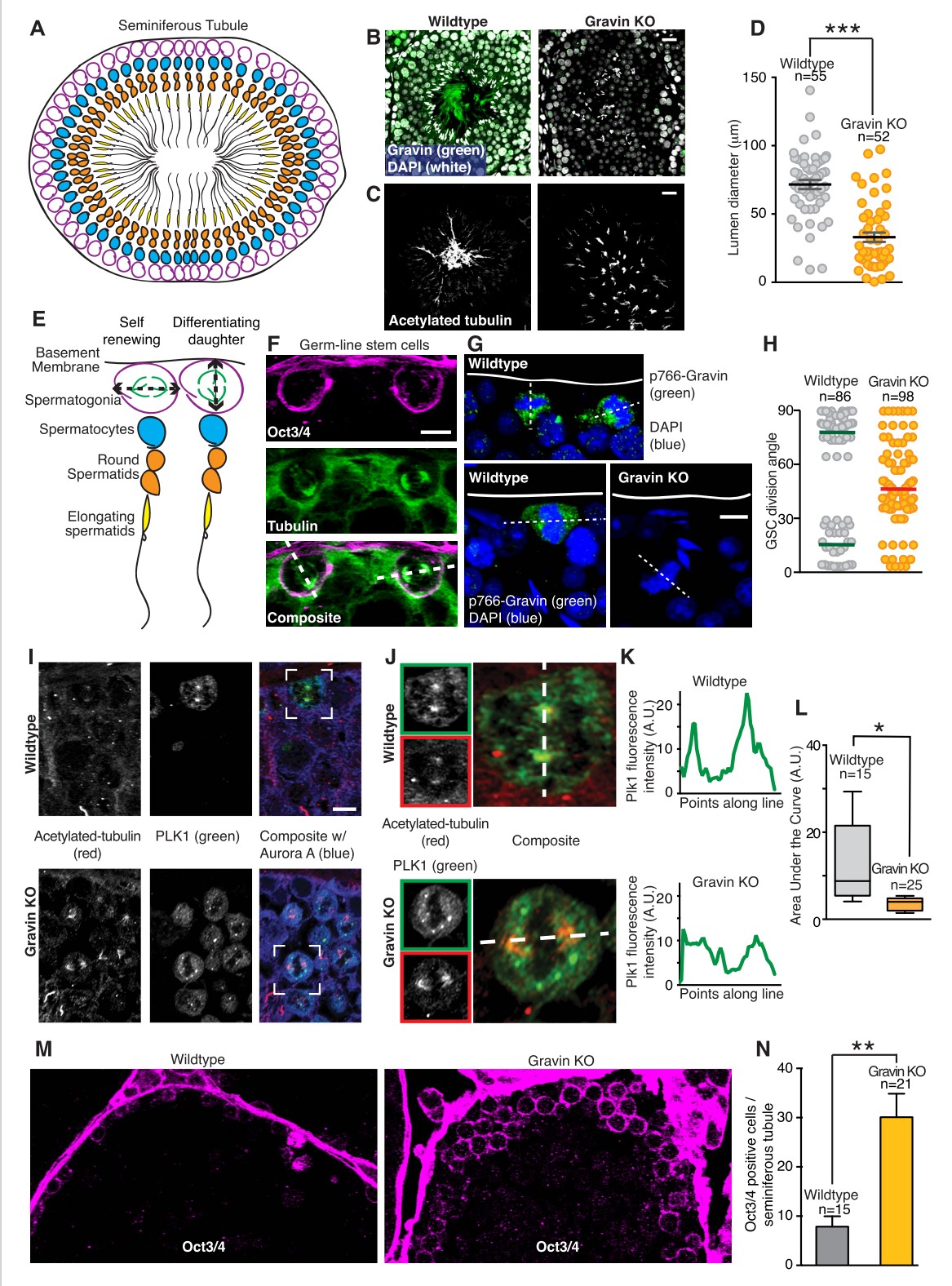

**Figure 8**. Gravin contributes to spindle orientation of germ line stem cells during spermatogenesis. (**A**) Schematic depicting the cross-sectional topology of a seminiferous tubule. The organization of the basement membrane (black), spermatogonia (magenta), spermatocytes (blue), round spermatids (orange), and elongating spermatids (yellow) is indicated. (**B**) Testis sections from (left) wild-type and (right) Gravin knockout mice stained for Gravin (green) and DAPI (white). (**C**) Parallel sections were stained for flagellum (acetylated tubulin) revealing a loss in polarized organization of seminiferous

*Figure 8. continued on next page*

*Figure 8. Continued*

tubules in Gravin knockout mice. Bar denotes 10 μm. (**D**) Lumen diameter was measured within seminiferous tubules of wild-type and Gravin knockout mice (total number of lumen measured are indicated on graph, and data are from three independent experiments, ***p < 0.001, ±SEM). (**E**) Close-up of model in **A** showing germ line stem cells (spermatogonia, magenta) can undergo either self-renewing (parallel spindle angle, left) or differentiating divisions (perpendicular spindle angle, right). (**F**) Representative cross-section of wild-type mouse seminiferous tubule. Germ line stem cells are stained for Oct3/4 (magenta) and microtubules (green). Dashed line in the composite image that bisects both spindle poles was used to determine spindle angle orientation in relation to basement membrane. Two distinct spindle angle orientations are evident. Bar, 5 μm. (**G**) Representative images of mouse seminiferous tubule sections stained for p766-Gravin (green) and DNA (DAPI, blue). (Top and bottom left) wild-type and (bottom right) Gravin knockout tissue. Dashed line bisects both spindle poles and was used to identify spindle angle orientation in relation to basement membrane. Bar, 5 μm. (**H**) Quantitation of spindle angle orientation in germ line stem cell sections from seminiferous tubule (examples presented in **F** and **G**). Spindle angles relative to basement membrane were identified for wild-type and Gravin knockout mitotic cells. Wild-type mitotic cells fall into two spindle angle populations between 0 and 30° (self-renewing) and between 60 and 90° (differentiating). However, Gravin null mitotic cells had a more randomized distribution of spindle angles. Total mitotic cell numbers in each genotype are indicated on graph. Experiments were conducted on tissue sections from 3 mice of each genotype. (**I**) Cross-sections of (top) wild-type and (bottom) Gravin knockout mouse seminiferous tubules were stained for acetylated tubulin (red, left) and Plk1 (green, middle). Composite images (right) are shown. White boxes identify the regions that are magnified in **J**. Bar, 5 μm. (**J–K**) Insets from (**I**) are shown for (top) wild-type and (bottom) Gravin knockout sections at higher magnification. Dashed white line in composite image identifies line-scan measured to determine the distribution of Plk1 intensity at each spindle pole. (**K**) Line plot graphs show integrated intensity values for Plk1 for (top) wild-type and (bottom) Gravin knockout dividing cells. (**L**) Amalgamated data presented as area under the curve from line scans of wild-type (n = 15 cells) and Gravin knockout mitotic cells (n = 25 cells). For Box and Whiskers plot, the box extends from the 25th to 75th percentiles using a standard method of computation via Prism software. The line in the middle of the box is plotted as the median. (**M**) Representative (left) wild-type and (right) Gravin knockout mouse seminiferous tubule sections stained for Oct3/4 (magenta) to identify germ line stem cells. (**N**) Relative abundance of Oct3/4-positive cells per seminiferous tubule section from (gray) wild-type and (orange) Gravin knockout mice. Number of seminiferous tubules analyzed per genotype is denoted on graph (±SEM, n = 3 mice each genotype, **p < 0.001).

During self-renewal, a mitotic cell chooses to orient its spindle either parallel or perpendicular to the basement membrane (*Lagos-Cabré and Moreno, 2008*; *Neumüller and Knoblich, 2009*). Self-renewing, parallel stem cell division produces two daughter stem cells, whereas perpendicular stem cell division yields one differentiating daughter cell and one stem cell ([*Siller and Doe, 2009*] *Figure 8E*). Basement membrane sections of the seminiferous tubules were identified with the stem cell marker Oct3/4 to investigate the angle of spindle orientation in germ line stem cells (*Figure 8F*, magenta). Mitotic spindle orientation was established by counterstaining with tubulin (*Figure 8F*, green). Indeed, two populations of dividing stem cells were observed: those with a mitotic spindle perpendicular (~90°) to the basement membrane and those with a mitotic spindle parallel (~10°) to the basement membrane (*Figure 8F*). Staining for p766-Gravin (green) and DNA (blue) confirmed that the phosphorylated anchoring protein was present in the few cells undergoing mitosis (*Figure 8G*, top and lower left panels). In contrast, mitotic cells from Gravin knockout mice displayed a randomized mitotic spindle orientation (*Figure 8G*, lower right panel). Importantly, this increased incidence of spindle angle randomization was evident in mitotic Gravin knockout germ line stem cells (average angle of 47.67°, n = 98; *Figure 8H*). These findings indicate that interrupting spindle polarity may adversely affect the development within seminiferous tubules in Gravin knockout mice.

On the basis of our cell-based and biochemical studies in *Figures 3, 4*, we postulated that the spindle misorientation phenotype observed in vivo is a consequence of displacing Plk1 from mitotic spindle poles. Therefore, we established the subcellular distribution of this kinase in seminiferous tubule sections from wild-type and Gravin knockout mice (*Figure 8I–L*). Acetylated tubulin was used as a marker for mitotic spindle poles (*Figure 8I*, left panels; red). In wild-type sections, the Plk1 signal was prominent at mitotic spindle poles (*Figure 8I*, top mid panel; green). Conversely, in Gravin knockout sections, the Plk1 signal was more disperse (*Figure 8I*, lower mid panel; green). Examination of representative cells from both genotypes at higher magnification best illustrates this phenomenon (*Figure 8J*). Line scan analysis detected an asymmetric distribution of Plk1 at one spindle pole in wild-type cells (*Figure 8K*, top panel). Conversely, a more uniform distribution of this kinase was detected in Gravin knockout cells (*Figure 8K*, lower panel). Quantitative analysis of multiple cells confirmed these findings further suggesting that Gravin functions to preferentially anchor Plk1 at one spindle pole in vivo (*Figure 8L*). Unfortunately, antibody compatibility issues precluded counter-staining with mother spindle pool markers.

Two possible outcomes can arise from the loss of Gravin expression in stem cells. Existing stem cell populations that undergo misoriented division may expand exponentially. Alternatively, germ line

stem cells may prematurely differentiate thereby diminishing the total stem cell population. In order to delineate between these two possibilities, we counted the number Oct3/4-positive cells in seminiferous tubule sections (*Figure 8M*). A 3.82-fold increase in Oct3/4-positive cells was observed in Gravin knockout sections (n = 4; *Figure 8M,N*). Thus, randomized spindle angles and the concomitant over-proliferation of germ line stem cells may underlie the pathological abnormalities observed in Gravin knockout seminiferous tubule organization as well as human seminoma (*Figures 1, 8*).

## Discussion

Aurora A and Plk1 are two kinases generally considered to act together to promote spindle pole maturation (*Hannak et al., 2001*; *Soung et al., 2009*; *Lee and Rhee, 2011*; *Joukov et al., 2014*; *Kong et al., 2014*). We have discovered an additional role for both kinases in the establishment of spindle orientation during mitotic progression. A key element of our finding is that both active kinases must be associated with the scaffolding protein Gravin to fulfill this ancillary function. By combining biochemical approaches (*Figure 2*) with quantitative super-resolution imaging and enzymology (*Figures 3, 4*), we have uncovered a unique kinase complex that assembles at the mother spindle pole during metaphase. Formation of a Gravin-Aurora A-Plk1 scaffold at this location may facilitate signal relay as depicted in *Figure 5F* to drive the spatial and temporal phosphorylation pattern of as yet unknown mitotic substrates.

Spindle maturation occurs during the transition from prophase to metaphase when dynein-mediated transport of PCM along the microtubules culminates in the formation of a mitotic spindle (*Purohit et al., 1999*; *Mahen and Venkitaraman, 2012*). Surprisingly, we observed an asymmetric distribution of the Gravin kinase scaffold at the mother spindle pole (*Figure 4* and *Figure 4—figure supplement 1*). Clustering of these enzymes at this location can either catalyze the efficient assembly of the PCM or alternatively coordinate the assembly of astral microtubules that orient the mitotic spindle. The super-resolution imaging analysis of Gravin null MEFs in *Figure 3F* argues strongly for the latter as these cells are deficient in astral microtubules but contain equal amounts of a canonical PCM protein, pericentrin ((*Doxsey et al., 1994*); *Figure 3E,G*). This notion is also consistent with the anomalous spindle tilt angles that were measured in cultured cells (*Figure 7*). These observations were further validated in vivo upon analysis of murine seminiferous tubule sections lacking Gravin (*Figure 8*).

One new concept that emerges from our studies is that Gravin-mediated clustering of Aurora A and Plk1 at the mother spindle pole provides a means to more precisely regulate symmetric cell division. We offer four lines of inquiry to support this new mechanism. First, data presented in *Figure 3F,G* indicate that loss of Gravin impacts the protrusion of astral microtubules. Second, data presented in *Figure 8* show that symmetric cell division is lost in Gravin null MEFs. Third, we measure increased mitotic spindle angles (>20°) in Gravin shRNA-treated cells compared to controls (*Figure 7A–E*). Fourth, a logical and important mechanistic extension of this latter observation is our finding that astral microtubules are lost from the mother spindle poles in Gravin null MEFs. Moreover, we postulate that the sequestering these enzymes at the mother spindle pole ensures that each kinase is optimally positioned to play a role in the regulation of astral microtubule protrusion, a process that influences the correct orientation of mitotic spindles during cell division. This mechanism is compatible with two recent reports indicating that signaling events downstream Plk1 modulate the correct orientation mitotic spindles (*Hanafusa et al., 2015*; *Yan et al., 2015*). However, a vital new piece of this puzzle, uncovered solely from our work, is that Gravin functions as the anchor for these enzymes.

Another theme emerging from the data in *Figure 8* is that the orientation of the mitotic spindle becomes increasingly important during stem cell division. These events influence cell fate determination and tissue organization (*Siller and Doe, 2009*; *Gillies and Cabernard, 2011*). In keeping with this notion, Gravin knockout mice exhibit an increase in Oct3/4, a marker for stem cells within the seminiferous tubule (*Figure 8M,N*). This is evocative of clinical studies that detect a marked increase in Oct3/4-positive cells in solid-state tumors including seminoma (*Singh et al., 2011*). These clinical results are entirely consistent with the data presented in *Figure 1* depicting a loss in tissue morphology observed in tissue sections from seminoma patients (*Figure 1C,D*). At the molecular level, we also detect decreased Gravin, Aurora A, and Plk1 levels in several samples collected from seminoma patients (*Figure 1A,B*). Therefore, we postulate that abnormalities in Gravin expression that promote mislocalization of Aurora A and Plk1 contributes to defects in orientation of mitotic

spindles to potentiate seminoma progression. However, Gravin knockout mice do not develop seminomas, yet some of the aforementioned abnormalities are evident in these animals. For example, these animals have difficultly breeding and upon aging display a tendency to develop prostate hyperplasia (*Akakura et al., 2008*). Both of these phenotypes may be indicative of developmental defects in polarized cell division. Therefore, we speculate that perturbation of Gravin-mediated signaling events during mitosis may promote a chronic decrease in organization of the seminiferous tubules with concomitant impairment of reproductive viability. Finally, when the mouse studies are considered in the context of our biochemical analysis and imaging of clinical samples, we can conclude that Gravin constrains Aurora A and Plk1 in an asymmetric manner to control spindle orientation during mitosis. Thus, we identify and define a new macromolecular signaling scaffold that drives stem cell maintenance and cell differentiation in seminiferous tubules.

## Materials and methods

### Reagents

The following antibodies were used: mouse α-tubulin, FITC-conjugated α-tubulin (Sigma, St. Louis, MO, United States), rabbit anti-pericentrin [M8; (*Doxsey et al., 1994*)], goat anti-Aurora A (Sigma), rabbit anti-p-Aurora A T288 (Cell Signaling Technology, Danvers, MA, United States), mouse anti-Aurora B (Abcam, Cambridge, MA, United States), human anti-CREST (Antibodies Incorporated, 15–234), rabbit anti-cenexin (Protein Tech Group, Chicago, IL, United States), rabbit p-Gravin T766A (*Canton et al., 2012*), mouse Gravin (Sigma; clone JP74), mouse anti-GAPDH (Sigma; GAPDH71.1), mouse anti-phospho-S10 Histone H3 (abcam; ab14955), rabbit anti-Plk1 (Cell Signaling Technology), mouse anti-Plk1 (Santa Cruz Biotechnology, Inc., Santa Cruz, CA, United States), mouse anti-Plk1 (Millipore; 35–206, Billerica, MA, United States), rabbit anti p-Plk1T210 (Cell Signaling Technology), mouse anti-Flag and Flag-HRP (Sigma), mouse anti-Par3 (Sigma), rabbit anti-Oct3/4 (Santa Cruz Biotechnology, Inc.), mouse anti-centrin (clone 20H5, EMD Millipore), mouse anti-acetylated tubulin (Sigma), mouse anti-centrobin (Abcam). Anti-rabbit, anti-mouse, and anti-goat HRP-conjugated secondary antibodies were purchased from GE Healthcare. Anti-rabbit, anti-mouse, and anti-goat secondary antibodies were purchased from Life technologies conjugated to Alexa Fluor 488, 647, and 568. DAPI (prolong anti-fade diamond, Invitrogen, Carlsbad, CA, United States) and phalloidin (Alexa Fluor 568, Invitrogen) were purchased from Life Technologies. The Gravin knockout (encoded by *Akap12* locus) mice were generated as described in (*Akakura et al., 2008*) and obtained from Irwin Gelman (Roswell Park Cancer Institute).

### Cell culture, transfection, and generation of stable Cell lines

Hela cells, U2OS, and MEFs (primary and immortalized) were maintained in D (Dulbecco's)-minimal essential medium (MEM) and retinal pigment epithelial cells (RPE) were maintained in DMEM:F12. All media was supplemented with 10% fetal bovine serum (FBS), 100 U/ml penicillin/streptomycin, and 1% Glut-MAX (Invitrogen). Infections for generation of stable knockdowns were performed with shRNA lentiviral particles (Santa Cruz Biotech) or retroviral particles (for immortalization). Transient gene expression was performed by transfection using *Trans*IT-LTI reagent (Mirus) for Hek293 cells, Hela monster (Mirus) for Hela cells, or by nucleofection using Ingenio (mirus) for RPE cells.

### Generation of MEFs

MEFs were isolated following the protocol provided by (*Chen et al., 2014*). Briefly, a timed pregnant female was sacrificed at embryonic day 12–13. Under sterile conditions, embryos were dissected from their placenta and surrounding membranes, and their organs and head were removed. Fibroblasts were isolated by trypsinization of minced tissue (0.25% trypsin in DMEM). Cells were grown in DMEM, 10% FBS, and penicillin/streptomycin at 37°C and used for immunofluorescence analysis immediately at passage 0–2. Immortalized MEF lines were established following standard protocols (*Chen et al., 1997*).

### Histological analysis

All human specimens were purchased from BioChain Institute, Inc. Reproductive age male mice (~7 weeks of age) were sacrificed, testes were removed, fixed in formalin for >24 hr at 4°, and embedded in paraffin. Samples were sectioned at 5 μm, mounted onto slides, and subjected to H&E or

conventional antigen retrieval through deparaffination followed by immunostaining. Sections were deparaffinized, rehydrated, and incubated with antibodies as labeled.

## Microscopy

### Spinning disk confocal microscopy

Images for spindle tilt, tissue sections, and general spindle morphology were acquired using primarily a Yokogawa CSU10 spinning disk mounted on a DM16000B inverted microscope (Leica, ×63 Plan-Apocromat NA 1.4 Oil Objective) with an Andor ILE laser launch with 50 mW Coherent OBIS lasers (405, 488, 561, and 642) unless otherwise noted in the manuscript. Two separate cameras were used depending on whether it was live-cell acquisition (Hamamatsu ImagEM EM-CCD Camera C9100-13) or fixed samples (CoolSnap HQ camera, Photometrics). Z-stacks were shown as 2D maximum projections or processed for 3-dimensional rendering (Metamorph). Fluorescence range intensity was adjusted identically for each series of panels. Intensity profiles and fluorescence intensity quantification were obtained from sum projections of Z stacks using either Metamorph or ImageJ/Fiji software. Fluorescence intensity quantification of spindle poles was carried out as previously described (*Chen et al., 2014*; *Hehnly and Doxsey, 2014*). In short, computer-generated concentric circles of 60 (inner area) or 80 (outer area) pixels in diameter were used to measure spindle pole (inner area) and calculate local background (difference between the outer and inner area) fluorescence intensity. Spindle angle measurements were carried out as previously described (*Chen et al., 2014*; *Hehnly and Doxsey, 2014*).

### GSDIM microscopy

Coverslips that were fixed and stained with primary antibodies towards Plk1, Aurora A, Cenexin, Centrobin, p-Gravin (T766A), and Gravin for 1 hr and followed with secondary antibodies (Alexa Fluor 647 or Alexa Fluor 568). Coverslips were mounted with MEA-GLOX imaging buffer (50 mM Tris pH 8.0, 10 mM NaCl, 0.56 mg/ml glucose oxidase, 34 µg/ml catalase, 10% wt/vol glucose, 100 mM MEA) on glass depression slides (neoLab, Heidelberg, Germany) and sealed with Twinsil (Picodent, Wipperfurth, Germany). Ground state depletion (GSD) super-resolution images of mitotic spindle poles were generated using a Leica SR GSD 3D system. The system is built around a Leica DMI6000 B TIRF microscope and is equipped with a Leica oil-immersion HC PL APO 160×/1.43 NA super-resolution objective, four laser lines (405/30 mW, 488 nm/300 mW, 532 nm/500 mW, and 642 nm/500 mW), and an Andor iXon3 EM-CCD. Images were collected in epifluorescent mode at a frame rate of 100 Hz for 50,000–100,000 frames using Leica Application Suite (LAS AF) software. Intensity calculations and 3-dimensional heatmaps were done in ImageJ/Fiji.

### SIM

Super-resolution 3D-SIM images were acquired on a DeltaVision OMX V4 (GE Healthcare) equipped with a 60×/1.42 NA PlanApo oil immersion lens (Olympus), 405-, 488-, 568-, and 642-nm solid-state lasers and sCMOS cameras (pco.edge). Image stacks of 5–6 µm with 0.125-µm thick z-sections and 15 images per optical slice (3 angles and 5 phases) which were acquired using immersion oil with a refractive index 1.518. Images were reconstructed using Wiener filter settings of 0.003 and optical transfer functions measured specifically for each channel with SoftWoRx 6.1.3 (GE Healthcare) to obtain super-resolution images with a twofold increase in resolution both axially and laterally. Images from different color channels were registered using parameters generated from a gold grid registration slide (GE Healthcare) and SoftWoRx 6.1.3 (GE Healthcare).

## Immunofluorescence

Staining procedures for cultured cells were performed as previously described (*Hehnly et al., 2006*; *Hehnly and Doxsey, 2014*).

## Immunoprecipitation, in vitro kinase assays, and immunoblotting

Procedures were performed as described in (*Canton et al., 2012*).

## Live-cell imaging

Cells were cultured on 35-mm dishes containing a central 14 mm 1.5 glass coverslip (MatTek). The cells were imaged in DMEM without phenol plus 20 mM HEPES (4-(2-hydroxyethyl)-1-piperazineethanesulfonic

acid) and 10% FBS at 37°C. Spinning disk confocal microscopy was performed on the above system attached to a Hamamatsu ImagEM, EM-CCD Camera C9100-13. For GFP imaging, frames were acquired every 2 to 5 min with an exposure time of 100 ms.

## Spindle orientation assay

Determination of spindle orientation relative to the horizontal plane or basement membrane in seminiferous tubules was performed as previously described (*Chen et al., 2014*; *Hehnly and Doxsey, 2014*). Briefly, spindle angle was estimated using inverse trigonometric functions, specifically, arctan (*Kuo et al., 2011*). Thus, if two spindle poles are in focus at the same z-plane, the estimated spindle orientation would be 0°. For cultured cells, at least 25 mitotic spindles were scored for each category in each experiment. For asymmetric divisions in the seminiferous tubules of the testis, a total of 5–10 tissue sections were analyzed per mouse. For each tissue sample, a z-series with depths of 5 μm (0.2 μm per step) were collected.

## PLA

Procedures were done as described in *Samelson et al. (2015)*.

## Statistical analysis

Statistics were performed using paired Student's *t*-test or unpaired with Welch's correction or Mann–Whitney *U*-test, for two-group comparisons using Prism's Graph Pad. p-values less than 0.05 were considered statistically significant.

## Acknowledgements

The work was supported by National Institutes of Health grants K99GM107355 (to HH), DK105542 and DK054441 (to JDS), and CA174527 and GM069429 (to LW). We are grateful to Claudia Moreno (UW), Patrina Pellett (GE Healthcare), and Tony Cooke (Leica) for technical and scientific help with super-resolution imaging techniques. We thank Juan-Jesus Vicente and Justin Decarreau (Wordeman Laboratory, UW) and Donelson Smith (Scott Laboratory, UW) for their assistance with live cell imaging and experimental design. We thank Mira Krendel (SUNY Upstate) for technical and experimental assistance. We thank Stephen Doxsey (UMMS) for his advice and guidance in mechanisms that govern spindle orientation.

## Additional information

### Funding

| Funder | Grant reference | Author |
| --- | --- | --- |
| National Institutes of Health (NIH) | DK105542 | John D Scott |
| National Institutes of Health (NIH) | CA174527 | Linda Wordeman |
| National Institutes of Health (NIH) | DK054441 | John D Scott |
| National Institutes of Health (NIH) | GM107355 | Heidi Hehnly |
| National Institutes of Health (NIH) | GM069429 | Linda Wordeman |
| Howard Hughes Medical Institute (HHMI) | | John D Scott |

The funders had no role in study design, data collection and interpretation, or the decision to submit the work for publication.

### Author contributions

HH, DC, Conception and design, Acquisition of data, Analysis and interpretation of data, Drafting or revising the article; PB, Acquisition of data, Analysis and interpretation of data; LKL, Drafting or revising the article; LO, Acquisition of data; IG, Contributed unpublished essential data or reagents; LFS, LW, Conception and design, Analysis and interpretation of data; JDS, Conception and design, Analysis and interpretation of data, Drafting or revising the article

## Ethics

Animal experimentation: This study was performed in strict accordance with the recommendations in the Guide for the Care and Use of Laboratory Animals of the National Institutes of Health. All of the animals were handled according to approved Institutional Animal Care and Use Committee (IACUC) protocols (#4196-01) of the University of Washington.

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
