## [Decision Letter]

Thank you for submitting your work entitled “A mitotic kinase scaffold depleted in testicular seminomas impacts spindle orientation in germline stem cells” for peer review at *eLife*. Your submission has been favorably evaluated by Fiona Watt (Senior editor) and three reviewers, one of whom is a member of our Board of Reviewing Editors.

The reviewers have discussed the reviews with one another and the Reviewing editor has drafted this decision to help you prepare a revised submission.

Hehnly et al. report that Gravin is required for proper spindle orientation in germ line stem cells during spermatogenesis and is depleted in seminomas. Using a combination of techniques in a variety of cell types, the authors have demonstrated that Gravin interacts with Aurora A and Plk1, and recruits these kinases at the mother spindle pole for proper assembly of astral microtubules. The data presented are interesting and the manuscript is well organized. However, the mechanisms by which Gravin scaffold accumulation at the mother spindle pole ensures proper spindle orientation in germline stem cells remain unclear.

1) The authors propose that Gravin ensures symmetric, but not asymmetric, cell division (Introduction). However, it is unclear whether Gravin is critical for successful symmetric or asymmetric stem cell division. First, Oct3/4 positive stem cells are increased in Gravin^−/−^ cells (Figure 6). As symmetric stem cell division produces two daughter stem cells (subsection “Gravin contributes to spindle orientation of germline stem cells during spermatogenesis”), defects in parallel symmetric cell division would lead to detachment of one daughter cell from the basal membrane and an expected decrease in stem cell population. Second, if Gravin is required for symmetric cell division, the authors need to propose a mechanistic model by which Gravin-mediated asymmetric accumulation of Aurora A and Plk1 at the mother spindle pole contributes to a parallel-symmetric division. As mentioned in the Introduction, the mother centrosome contributes to the perpendicular asymmetric cell division in Drosophila germ line stem cell by linking the mother centrosome to the stem cell niche (55).

2) The authors demonstrate that there are two populations of dividing germ line stem cells in Figure 6. It is critical to analyze which populations contain phosphorylated Gravin, although Figure 6 shows that p766-Gravin signals are detected in symmetrically dividing self-renewing cells. In addition, it is important to analyze if p766-Gravin accumulates at the mother spindle pole in the germ line stem cells similar to MEFs.

3) The authors demonstrate that p766-Gravin, Aurora A and Plk1 accumulate at the mother spindle pole in MEFs (Figure 3). However, the exact contribution of spindle pole asymmetry to astral microtubule formation is still unclear. The authors need to compare the astral microtubules derived from mother and daughter spindle poles, respectively, as shown in Figure 3.

4) The authors conclude that Gravin is important for mitotic progression. Gravin^−/−^ MEF grew more slowly than wt MEF. In contrast, analysts of seminiferous tubule sections demonstrated cell with increased mitotic index. The authors need to clarify these observations – do both cell types grow slowly in the absence of gravin and the increased mitotic index simply a reflection of a mitotic delay. Or does Gravin have opposite phenotypes in the cells?

5) The immunoblot analysis of tumors (Figure 1) shows loss of Gravin, but also loss of Plk1 and AuroraA. Are these related events? Do Gravin^−/−^ MEF show decreased Plk1 and AuroraA? If so, does the loss of total protein explain the decreased spindle pole localization? Does loss of protein expression account for the role of Gravin to mediate interactions between Plk1 and AuroraA.

6) The rescue experiments using the T766A mutant of Gravin should be extended to demonstrate that it is unable to recruit Plk1 to the mother spindle pole to regulate astral microtubules.

7) The authors state data as xxx {plus minus} yyy in several places in the text. The numbers should be defined. Is xxx a mean or a median or something else? Is yyy the SD, SEM or something else? Also, do the statistics justify the precision of the numbers provided – for example, 64.57 {plus minus} 4.09 (in the first paragraph of subsection “Gravin anchoring of Aurora A and Plk1 is required for mitotic progression”).

---

## [Author Response]

*1) The authors propose that Gravin ensures symmetric, but not asymmetric, cell division (Introduction). However, it is unclear whether Gravin is critical for successful symmetric or asymmetric stem cell division. First, Oct3/4 positive stem cells are increased in Gravin*^*−/−*^
*cells (*Figure 6*). As symmetric stem cell division produces two daughter stem cells (subsection “Gravin contributes to spindle orientation of germline stem cells during spermatogenesis”), defects in parallel symmetric cell division would lead to detachment of one daughter cell from the basal membrane and an expected decrease in stem cell population*.

We apologize for the confused explanation in the original manuscript. Our contention is that loss of Gravin favors unregulated spindle orientation. This promotes an increased likelihood of randomized cell division (Figure 8; orange points). This concept is evident upon the analysis of germline stem cell populations in seminiferous tubule tissue sections from wildtype and *Gravin*^*−/−*^ mice. In wildtype cells quantitative analysis of tissue sections stained for tubulin reveals two distinct populations of spindle organization. In contrast, we detect a randomized spindle orientation in Gravin null tissue (Figure 8). On the basis of these results we reason that loss of Gravin induces a higher incidence of cells that retain a stem cell phenotype.

In addition we have performed new experiments showing that that division angles are randomized in Gravin^−/−^ MEFs as compared to wildtype controls (Figure 7, and Video 2). These videos underscore the quantitation of spindle angles measured in control and Gravin shRNA treated cells as depicted in Figure 7. These new data are included in Figure 7, and Video 2 has been added as supplemental material. These data are discussed in subsection “The Gravin macromolecular complex is required for spindle orientation.”

*Second, if Gravin is required for symmetric cell division, the authors need to propose a mechanistic model by which Gravin-mediated asymmetric accumulation of Aurora A and Plk1 at the mother spindle pole contributes to a parallel-symmetric division. As mentioned in the Introduction, the mother centrosome contributes to the perpendicular asymmetric cell division in Drosophila germ line stem cell by linking the mother centrosome to the stem cell niche (*[55]*)*.

This is a valid point. There are now four lines of inquiry to support our notion that the Gravin kinase scaffold contributes to symmetric cell division. First, data presented in Figure 3 indicate that loss of Gravin impacts the protrusion of astral microtubules. Second, data added in Figure 7 and Video 2 show that symmetric cell division, is lost in Gravin^−/−^ MEFs. Third, we measure increased mitotic spindle angles (> 20°) in Gravin shRNA treated cells as compared to controls (Figure 7). Fourth, a logical and mechanistic extension of this latter observation was to establish if astral microtubules are preferentially lost from the mother spindle poles in Gravin^−/−^ MEFs. New studies presented Figure 4 measure a reduction in astral microtubule length from the mother spindle pole as compared to analogous structures protruding from the daughter spindle pole in these cells.

Thus, our combined biochemical, cellular and super-resolution imaging strategy allows us to propose that Gravin preferentially anchors two mitotic kinases, Plk1 and Aurora A, at the mother spindle pole. Moreover, we postulate that the sequestering of these enzymes at the mother spindle pole ensures that each kinase is optimally positioned to play a role in the regulation of astral microtubule protrusion, a process that influences the correct orientation of mitotic spindles during cell division.

This mechanism is also consistent with two recent reports (now cited in the revised manuscript) indicating that signaling events downstream of Plk1 modulate the correct orientation of mitotic spindles. However a vital new piece of this puzzle, solely uncovered from our work, is that Gravin functions as the anchor for these enzymes. Interpretation of the new data in Figure 4 is introduced to the Results section. A succinct discourse of our proposed mechanism for Gravin action in “symmetric cell division” has been incorporated into the Discussion.

*2) The authors demonstrate that there are two populations of dividing germ line stem cells in*
Figure 6*. It is critical to analyze which populations contain phosphorylated Gravin, although*
Figure 6
*shows that p766-Gravin signals are detected in symmetrically dividing self-renewing cells. In addition, it is important to analyze if p766-Gravin accumulates at the mother spindle pole in the germ line stem cells similar to MEFs*.

We have responded to the reviewers' suggestions in three ways: i) p766-Gravin is present in all germline stem cells and wildtype MEFs during mitosis, irrespective of which direction either cell-type divides. To emphasize this point we have included an additional representative image of p766-Gravin in Figure 8 and modified the text (please see the subsection “Gravin contributes to spindle orientation of germline stem cells during spermatogenesis”).

ii) Perhaps not surprisingly, the asymmetric distribution of p766-Gravin between the mother and daughter spindle poles is most evident upon analysis by super resolution microscopy. In keeping with this statement we now include a comparison between data collected by conventional wide field microscopy and structured illumination microscopy in Figure 2—figure supplement 1. Likewise ground state depletion microscopy provides a much better resolution as depicted in Figure 4—figure supplement 1.

iii) Finally, we wish to point out that super resolution microscopy is technically challenging and often prohibitive when attempted on paraffin embedded tissue samples such as the sections used in Figure 8. Therefore a more feasible approach was to conduct super resolution analysis in wildtype and Gravin^−/−^ MEFs. We contend these cell lines represent a valid model system that recapitulates elements of the in vivo environment. Particularly since we observe the same randomization of mitotic spindle angles upon loss or depletion of Gravin in all systems (Figure 7, new Video 2 and Figure 8). This new information is discussed in subsection “The Gravin macromolecular complex is required for spindle orientation”.

*3) The authors demonstrate that p766-Gravin, Aurora A and Plk1 accumulate at the mother spindle pole in MEFs (*Figure 3*). However, the exact contribution of spindle pole asymmetry to astral microtubule formation is still unclear. The authors need to compare the astral microtubules derived from mother and daughter spindle poles, respectively, as shown in*
Figure 3.

Thank you for this suggestion, we have quantified the astral microtubule length and number at each spindle pole in wildtype and Gravin^−/−^ cells (Figure 4). Using nanoscopic microscopy (SIM) techniques we detect a marked decrease in the number and length of astral microtubules at the mother spindle pole in Gravin^−/−^ cells. Conversely, there is minimal change in astral microtubules at the daughter spindle pole. These new findings are presented in Figure 4 and Figure 4—figure supplement 1. These new data are discussed in the subsection “Clustering Aurora A and Plk1 at spindle poles requires Gravin”.

*4) The authors conclude that Gravin is important for mitotic progression. Gravin*^*−/−*^
*MEF grew more slowly than wt MEF. In contrast, analysts of seminiferous tubule sections demonstrated cell with increased mitotic index. The authors need to clarify these observations – do both cell types grow slowly in the absence of gravin and the increased mitotic index simply a reflection of a mitotic delay. Or does Gravin have opposite phenotypes in the cells?*

We have added additional data to address this point. Time-lapse video microscopy of Gravin shRNA treated cells reveal a delayed progression through mitosis when compared to cells treated with shRNA controls. This information is now included as Video 1. This new live cell imaging data is consistent with an increased mitotic index upon the depletion of Gravin (7). These findings are also comparable to the increase in the mitotic index measured in murine Gravin^−/−^ seminiferous tubules (Figure 1).

*5) The immunoblot analysis of tumors (*Figure 1*) shows loss of Gravin, but also loss of Plk1 and AuroraA. Are these related events? Do Gravin*^*−/−*^
*MEF show decreased Plk1 and AuroraA? If so, does the loss of total protein explain the decreased spindle pole localization? Does loss of protein expression account for the role of Gravin to mediate interactions between Plk1 and AuroraA.*

Immunoblot data of clinical samples from seminoma patients presented in Figure 1 detect reduced levels of Gravin and concomitant reductions in the signals for Aurora A and Plk1. In an attempt to respond to the reviewers' request we further explore this phenomena in mouse embryonic fibroblasts (MEFs) and human HEK293 cells. These experiments were only partially successful.

Despite eleven independent attempts, immunobot detection of the murine Plk1 and Aurora A orthologs in MEFs was sporadic. A majority of the time we were unable to detect either kinase by immunoblot. Yet on the few occasions when we were able to detect any Plk1 signal in mitotic Gravin^−/−^ MEFs the data were inconclusive. The Plk1 immunoblot signal was reduced once, constant in one instance, and slightly increased in the final attempts. Thus, we were forced to conclude that although the Aurora A and Plk1 antibodies are reliable for immunofluorescent detection of the human and murine kinases, they are less cross reactive with murine orthologs when used for immunoblot detection. As a backup approach we monitored the levels of the human Aurora A and Plk1 upon shRNA deletion of Gravin in HEK293 cells.

New data shows that gene silencing of Gravin had little or no effect on the levels of Aurora A or Plk1 during mitosis (Figure 4—figure supplement 1, lane 2, middle panels). In addition we have modified the text stating: “Additional control experiments in HEK293 cells revealed that cellular levels of both kinases were unaltered in absence of Gravin. This latter observation argues that ablation of Gravin in tissue culture cell lines instigates the displacement of both kinases from the spindle pole but may not affect the cellular levels of each enzyme. However, on the basis of our analysis of clinical samples in Figure 1 we propose that unidentified mitigating factors contribute to the mislocalization and depletion of both kinases in human tissue”.

*6) The rescue experiments using the T766A mutant of Gravin should be extended to demonstrate that it is unable to recruit Plk1 to the mother spindle pole to regulate astral microtubules*.

We have previously shown that the Gravin T766A mutant is unable to bind Plk1 in a variety of cell types. This information has been added to the text and the appropriate papers citied. In addition new data presented in Figure 4—figure supplement 1 demonstrates that expression of the Plk1 binding mutant Gravin T766A in Gravin^−/−^ MEFs is unable to rescue the number of astral microtubules protruding form the mitotic spindle. Rather, rescue with this Plk1-anchoring defective mutant may exacerbate the astral microtubule phenotype. These data are included in Figure 4—figure supplement 1 and discussed in the subsection “Clustering Aurora A and Plk1 at spindle poles requires Gravin”.

*7) The authors state data as xxx {plus minus} yyy in several places in the text. The numbers should be defined. Is xxx a mean or a median or something else? Is yyy the SD, SEM or something else? Also, do the statistics justify the precision of the numbers provided – for example, 64.57 {plus minus} 4.09 (in the first paragraph of subsection “Gravin anchoring of Aurora A and Plk1 is required for mitotic progression”)*.

We have clarified these points. In all cases where a bar graph is presented or where a scatter plot is presented XXX represents the mean and the +/- demonstrates the range in standard error of the mean. We have checked all the exact means added to the text and made sure they are consistent with the data presented. In the cases where we present a Box and Whiskers plot, the box extends from the 25th to 75th percentiles using a standard method of computation via Prism software. The line in the middle of the box is plotted as the median.